# MotionGPT: Human Motion as a Foreign Language

**Biao Jiang**[1,2*]     **Xin Chen**[2*]     **Wen Liu**[2]     **Jingyi Yu**[3]     **Gang Yu**[2]     **Tao Chen**[1†]

[1]Fudan University          [2]Tencent          [3]ShanghaiTech University

https://github.com/OpenMotionLab/MotionGPT

## Abstract

Though the advancement of pre-trained large language models unfolds, the exploration of building a unified model for language and other multimodal data, such as motion, remains challenging and untouched so far. Fortunately, human motion displays a semantic coupling akin to human language, often perceived as a form of body language. By fusing language data with large-scale motion models, motion-language pre-training that can enhance the performance of motion-related tasks becomes feasible. Driven by this insight, we propose MotionGPT, a unified, versatile, and user-friendly motion-language model to handle multiple motion-relevant tasks. Specifically, we employ the discrete vector quantization for human motion and transfer 3D motion into motion tokens, similar to the generation process of word tokens. Building upon this "motion vocabulary", we perform language modeling on both motion and text in a unified manner, treating human motion as a specific language. Moreover, inspired by prompt learning, we pre-train MotionGPT with a mixture of motion-language data and fine-tune it on prompt-based question-and-answer tasks. Extensive experiments demonstrate that MotionGPT achieves state-of-the-art performances on multiple motion tasks including text-driven motion generation, motion captioning, motion prediction, and motion in-between.

## 1 Introduction

Recent years have witnessed a significant breakthrough in pre-trained large language models such as GPT [34, 35, 3, 27], BERT [7], and T5 [36, 5], which lead to the convergence of language [59, 47], image [33, 50, 20], mesh [55, 26] and mutlimodal [8] modeling. Nevertheless, a general pre-trained model for human motion and language has yet to emerge. This pre-trained motion-language model, capable of supporting numerous motion-relevant tasks through prompts, should benefit diverse fields like gaming, robotics, virtual assistant, and human behavior analysis.

Previous research on human motion has explored various tasks, including motion generation [29, 10, 46, 52, 57], motion captioning [9, 11], and motion prediction [56, 61, 24]. Recent text-to-motion works[46, 58, 30, 52] have attempted to employ pre-trained language-relevant models [7, 33]. For instance, MDM [46] learns a motion diffusion model with conditional text tokens from CLIP [33], while MLD [52] integrates motion latent space to improve the efficiency of motion diffusion process. On the other hand, MotionCLIP [45] and TM2T [11] concentrate on modeling the coupled relationship between motion and text description. However, the above approaches treat motion and language as separate modalities, which often require strictly paired motion and text data. Moreover, since the supervisions are task-specific, they can hardly generalize effectively to unseen tasks or data, as they lack a comprehensive understanding of the relationship between motion and language. We thus focus on building a pre-trained motion-language model, which can generalize to various tasks and learn in-depth motion-language correlation knowledge from more feasible motion and language data.

Two challenges are crucial and need to be solved for pre-training a promising motion-language model. The first is modeling the relation between language and motion, and the second is building a uniform multi-task framework that can generalize to new tasks. Fortunately, human motion exhibits a semantic

---

[*]Contributed equally and work done while Biao Jiang was a Research Intern with Tencent PCG
[†]Corresponding author.

37th Conference on Neural Information Processing Systems (NeurIPS 2023).

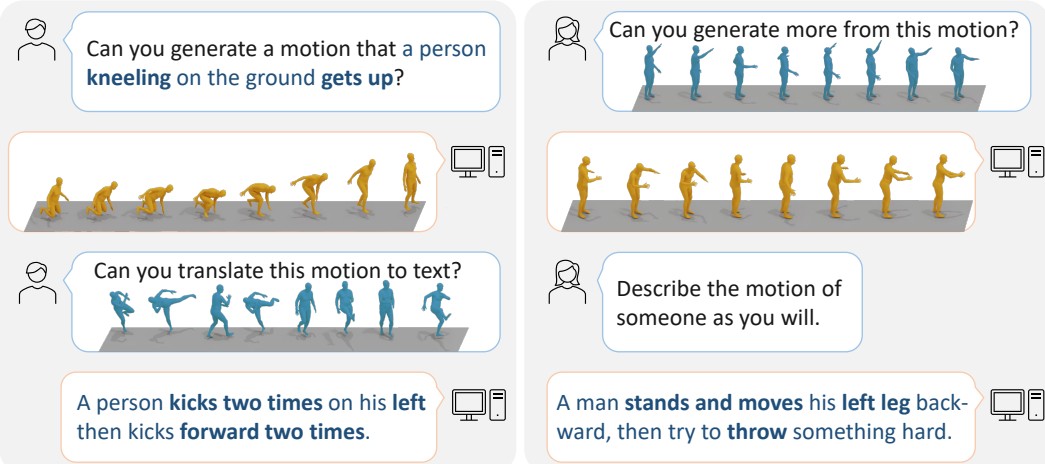

Figure 1: MotionGPT can address diverse motion-relevant tasks uniformly given different instructions. We provide the results on text-to-motion (the upper left), motion captioning (the bottom left), motion completion (the upper right), and the language question-to-answer (the bottom right). The left to right of motion represents the time order. Blue motion denotes the input, and yellow is the generation.

coupling similar to human language, often interpreted as a form of body language. Building upon this observation,we follow vision-language pre-training from BEiT-3 [50] to treat human motion as a specific foreign language. By integrating motion and language data together and encoding them within a single vocabulary, the relationship between motion and language becomes more apparent. Therefore, with recent significantly larger-scale language data and models, the motion-language pre-training has great potential to improve the performance on motion tasks. Meanwhile, this pre-training on language enables textual instructions like prompts in InstructGPT [27] and makes the model more versatile and user-friendly for various motion tasks.

In this work, we propose a uniform motion-language framework, namely MotionGPT, that leverages the strong language generation and zero-shot transfer abilities of pre-trained language models for doing human motion-related tasks. To enable MotionGPT to comprehend and generate human-like motions, we first learn a motion-specific vector quantized variational autoencoder (VQ-VAE) model to construct "motion vocabulary", akin to English vocabulary and then convert raw motion data into a sequence of motion tokens. These tokens are then processed by a pre-trained language model [36, 5] that learns the underlying grammar and syntax of the motion language, as well as its relationship with the corresponding textual descriptions. To effectively integrate language and motion in MotionGPT, we design a two-stage training scheme. We first pre-train the language model on the raw motion dataset to learn the basic grammar and syntax of the motion language. For prompt tuning, we fine-tune the language model on an instruction dataset, which contains both textual descriptions and motion data, to learn the correlation between the two modalities. Extensive experiments demonstrate that MotionGPT achieves state-of-the-art performance on text-to-motion, motion-to-text, motion prediction, and motion in-between.

We summarize our contributions as follows: (1) We propose a uniform motion-language generative pre-trained model, MotionGPT, which treats human motion as a foreign language, introduces natural language models into motion-relevant generation, and performs diverse motion tasks with a single model. (2) We introduce a motion-language training scheme with instruction tuning, to learn from task feedback and produce promising results through prompts. (3) We propose a general motion benchmark for multi-task evaluation, wherein MotionGPT achieves competitive performance across diverse tasks, including text-to-motion, motion-to-text, motion prediction, and motion in-between, with all available codes and data.

## 2 Related Work

**Human Motion Synthesis** involves generating diverse and realistic human-like motion using multi-modal inputs, such as text [10, 30, 58, 46, 11, 1, 17], action [29, 12, 46, 52], and incomplete motion [56, 61, 24, 46]. Text-to-motion is one of the most important motion generation tasks, due to the user-

| Methods | Text-to-Motion | Motion-to-Text | Motion Prediction | Motion In-between | Random Motion | Random Description |
|---|---|---|---|---|---|---|
| T2M-GPT [57] | ✔ | ✗ | ✔ | ✗ | ✔ | ✗ |
| MLD [52] | ✔ | ✗ | ✗ | ✗ | ✔ | ✗ |
| TM2T [11] | ✔ | ✔ | ✗ | ✗ | ✗ | ✗ |
| MDM [46] | ✔ | ✗ | ✔ | ✔ | ✔ | ✗ |
| MotionDiffuse[58] | ✔ | ✗ | ✔ | ✔ | ✔ | ✗ |
| MotionGPT (Ours) | ✔ | ✔ | ✔ | ✔ | ✔ | ✔ |

Table 1: Comparison of recent state-of-the-art methods on diverse motion-relevant tasks. *Random Motion* and *Random Caption* represent unconstrained generation of motions and motion descriptions.

friendly and convenient language input. MDM [46] proposes a diffusion-based generative model [14] separately trained on several motion tasks. MLD [52] advances the latent diffusion model [43, 38] to generate motions based on different conditional inputs. T2M-GPT [57] investigates a generative framework based on VQ-VAE and Generative Pre-trained Transformer (GPT) for motion generation. Motion completion task generates motion conditioning on partial motions, such as classical motion prediction [56, 61, 24] or motion in-between [46], which generates the intermediate motion while the first and last parts are fixed. Although they show promising results in various human motion tasks, most above methods are limited in using a single model to handle multiple tasks. We thus propose a uniform approach that treats human motion as a foreign language, and leverages the strong language generation and zero-shot transfer abilities of pre-trained language models

**Human Motion Captioning.** To describe human motion with natural languages, [44] learns the mapping from motions to language relying on two statistical models. Furthermore, recurrent networks have also been used in [54, 32]. More recently, TM2T [11] proposed a new motion representation that compresses motions into a short sequence of discrete variables, then uses a neural translation network to build mappings between two modalities. While previous research like TM2T [11] incorporated captioning modules into their training pipeline for motion generation, these approaches are constrained to bidirectional translation between text and motion within one uniform framework.

**Language Models and Multi-Modal.** Large-scale language models (LLMs) [7, 6, 36, 3, 59, 47], enabled by extensive datasets and model size, have demonstrated impressive comprehension and generation capabilities, elevating natural language processing to new heights. BERT [7] pre-trains deep bidirectional language representations that can support downstream tasks. T5 [36] introduced a unified framework that converts all text-based language problems into a text-to-text format. More recent research [51, 2, 27, 5] find that by fine-tuning pre-trained models using input-output pairs consisting of instructions and coupled answers, the performance of pre-trained models can be further improved. FLAN [5] presents an instruction-tuning technique that surpasses the performance of non-tuned models in unseen tasks. Recently, the wave of multi-modal models [20, 15, 19] is intriguing to process text along with other modalities, such as images [20, 15, 8], audio [13, 8], and videos [53]. CLIP [33] further learns a semantic latent representation that couples images with corresponding language descriptions. Despite the success of language models in various vision-language tasks, the development of multi-modal language models that can handle human motion is still limited.

**Motion Language Pre-training.** Existing text-to-motion generation methods [10, 30, 46, 11, 1, 17] can be characterized as caption-to-motion, where the models take in a pure text description of the desired motion. While these methods can generate motions from textual descriptions, they are often limited in supporting instructions from users like InstructGPT [27]. In other words, they do not allow users to provide context-specific instructions for certain applications. MotionCLIP [45] utilizes the language and visual understanding of CLIP [33] to align its latent space with a motion auto-encoder. Meanwhile, many language models, such as T5[36] and InstructGPT [27], have been developed to address diverse language processing tasks, including translation, question answering, and classification. These models are typically designed to map a given text input to a target output, such as a translation or answer. However, while these models have shown remarkable performance in language tasks, they have not been widely applied to motion tasks. Therefore, we propose MotionGPT to enable the effective integration of natural language models with human motion tasks, providing a unified solution for motion synthesis problems.

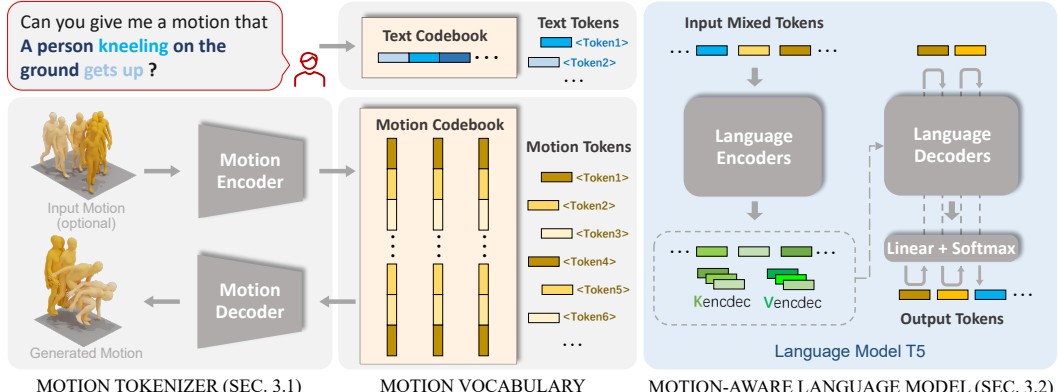

Figure 2: Method overview: MotionGPT consists of a motion tokenizer $\mathcal{V}$ (Sec. 3.1) and a motion-aware language model (Sec. 3.2). Combining *Motion Tokens* learned by $\mathcal{V}$ and *Text Tokens* by text tokenizer, we then learn motion and language jointly utilizing language model as backbone.

# 3 Method

To involve large language data and models in the motion generation tasks, we propose a unified motion-language framework named MotionGPT. As illustrated in Fig. 2, MotionGPT consists of a motion tokenizer responsible for converting raw motion data into discrete motion tokens (Sec. 3.1), as well as a motion-aware language model that learns to understand the motion tokens from large language pre-training models by corresponding textual descriptions (Sec. 3.2). To address motion-relevant tasks, we introduce a three-stage training scheme (Sec. 3.3) of MotionGPT for the training of motion tokenizer, motion-language pre-training, and instruction tuning.

We first propose the motion tokenizer consisting of a motion encoder $\mathcal{E}$ and a motion decoder $\mathcal{D}$, to encode a $M$ frame motion $m^{1:M} = \{x^i\}_{i=1}^M$ into $L$ motion tokens $z^{1:L} = \{z^i\}_{i=1}^L$, $L = M/l$, and decode $z^{1:L}$ back into the motion $\hat{m}^{1:M} = \mathcal{D}(z^{1:L}) = \mathcal{D}(\mathcal{E}(m^{1:M}))$, where $l$ denotes the temporal downsampling rate on motion length. Then, given an $N$ length sentence $w^{1:N} = \{w^i\}_{i=1}^N$ describing a motion-related question or demand, MotionGPT aims to generate its answer as $L$ length tokens $\hat{x}^{1:L} = \{\hat{x}^i\}_{i=1}^L$. It could be the human motion tokens $\hat{x}_m^{1:L}$ or the text tokens $\hat{x}_t^{1:L}$, which results in a motion $\hat{m}^{1:M}$ or a sentence $\hat{w}^{1:L}$ like a description of the given motion.

## 3.1 Motion Tokenizer

To represent motion in discrete tokens, we pre-train a 3D human motion tokenizer $\mathcal{V}$ based on the Vector Quantized Variational Autoencoders (VQ-VAE) architecture used in [48, 42, 11, 57]. Our motion tokenizer consists of an encoder $\mathcal{E}$ and a decoder $\mathcal{D}$. The encoder generates discrete motion tokens with high informative density, while the decoder is able to reconstruct the motion tokens into motion sequences $\hat{m}^{1:M}$. This approach enables us to efficiently represent motion as a language, facilitating the integration of motion and language for various motion-related tasks.

Specifically, the motion encoder $\mathcal{E}$ first applies 1D convolutions to given frame-wise motion features $m^{1:M}$ along the time dimension, to obtain latent vectors $\hat{z}^{1:L} = \mathcal{E}(m^{1:M})$. Next, we transform $\hat{z}$ into a collection of codebook entries $z$ through discrete quantization. The learnable codebook $Z = \{z^i\}_{i=1}^K \subset \mathbb{R}^d$ consists of $K$ latent embedding vectors, each of dimension $d$. The process of quantization $Q(\cdot)$ replaces each row vector $b$ with its nearest codebook entry $b_k$ in $Z$, written as

$$z_i = Q(\hat{z}^i) := \arg\min_{z_k \in Z} \|\hat{z}_i - z_k\|_2. \tag{1}$$

After quantization, the motion decoder $D$ project $z^{1:L} = \{z^i\}_{i=1}^L$ back to raw motion space as the motion $\hat{m}^{1:M}$ with $M$ frames. To train this motion tokenizer, we follow [11, 57] to utilize three distinct loss functions for training and optimizing the motion tokenizer: $\mathcal{L}_{\mathcal{V}} = \mathcal{L}_r + \mathcal{L}_e + \mathcal{L}_c$, where the reconstruction loss $\mathcal{L}_r$, the embedding loss $\mathcal{L}_e$, and the commitment loss $\mathcal{L}_c$. To further improve the generated motion quality, we follow [57] to utilize L1 smooth loss and velocity regularization in the reconstruction loss, as well as exponential moving average (EMA) and codebook reset techniques [37]

to enhance codebook utilization during training. We provide more details about the architecture and the training of our motion tokenizer in the supplement.

## 3.2 Motion-aware Language Model

Employing this motion tokenizer, a human motion $m^{1:M}$ can be mapped to a sequence of motion tokens $z^{1:L}$, allowing for joint representation with similar vocabulary embedding in language models [18, 36, 27]. By combining them in the unified vocabulary, we then learn motion and language jointly. We first represent motion tokens $z^{1:L}$ as a sequence of indices $s^{1:L} = \{s^i\}_{i=1}^L$, where $s^i$ corresponds to the index number of motion tokens $z^{1:L}$. On the other hand, previous language models, such as T5 [36], encode text as WordPiece tokens. They utilized a vocabulary of $K_t$ word pieces and trained the SentencePiece [18] model on a mixture of language datasets.

Most previous text-to-motion [11, 52, 57] or motion-to-text [11] approaches employ different modules to handle text and motion individually, while we aim to model text and human motion together and in the same way. To achieve this, we combine the original text vocabulary $V_t = \{v_t^i\}_{i=1}^{K_t}$ with motion vocabulary $V_m = \{v_m^i\}_{i=1}^{K_m}$, which is order-preserving to our motion codebook $Z$. Moreover, $V_m$ includes several special tokens like boundary indicators, for example, </som> and </eom> as the start and end of the motion. Thus, we employ a new unified text-motion vocabulary $V = \{V_t, V_m\}$, and can formulate diverse motion-related tasks in a general format, where both input "words" and output "words" are from the same $V$. These "words" can represent natural language, human motion, or even a mixture of two, depending on the specific task to be solved. Therefore, our MotionGPT allows for the flexible representation and generation of diverse motion-related outputs within a single model.

To address the conditioned generation task, we employ a transformer-based model based on the architecture proposed in [36], which effectively maps the input sequences to the output. Our source input consists of a sequence of tokens $X_s = \{x_s{}^i\}_{i=1}^N$, where $x_s \in V$ and $N$ represents the input length. Similarly, the target output is $X_t = \{x_t{}^i\}_{i=1}^L$, where $x_t \in V$ and $L$ denotes the output length. As shown in Fig. 2, the source tokens are fed into the transformer encoder, and the subsequent decoder predicts the probability distribution of the potential next token at each step $p_\theta(x_t \mid x_s) = \prod_i p_\theta \left( x_t^i \mid x_t^{<i}, x_s \right)$ in an autoregressive manner. Therefore, during the training process, the objective is to maximize the log-likelihood of the data distribution:

$$\mathcal{L}_{LM} = - \sum_{i=0}^{L_t-1} \log p_\theta \left( x_t^i \mid x_t^{<i}, x_s \right). \tag{2}$$

By optimizing this objective, MotionGPT learns to capture the underlying patterns and relationships from the data distribution, facilitating the accurate and meaningful generation of the target "words". During the inference process, the target tokens are sampled recursively from the predicted distribution $p_\theta \left( \hat{x_t}^i \mid \hat{x_t}^{<i}, x_s \right)$ until the end token (i.e., ). This sampling strategy enables the generation of the target sequence in a step-by-step manner, where each token is probabilistically determined based on the previously generated tokens and the given source input.

## 3.3 Training Strategy

Since T5s have only been exposed to language data, represented within a text vocabulary $V_t$, we thus bridge motion and language and enable this language model to comprehend human motion concepts, by learning the motion vocabulary $V_m$. As shown in Fig. 3, our training scheme includes three stages: (1) Training of motion tokenizer, which focuses on learning the motion codebook to represent human motion as discrete tokens. (2) Motion-language pre-training stage, which includes unsupervised and supervised objectives to learn the relationship between motion and language. (3) Instruction tuning stage, which tunes the model based on prompt-based instructions for different motion-relevant tasks.

**Training of Motion Tokenizer.** We first learn the motion tokenizer using the objective defined in Equation 3.1. This training process allows any human motion sequence $\hat{x}^{1:L}$ to be represented as a sequence of motion tokens, enabling seamless integration with textual information. Once optimized, the motion tokenizer remains unchanged throughout the subsequent stages of the pipeline.

**Motion-language Pre-training Stage.** The T5 models [36, 5] are trained and fine-tuned on natural language datasets with instruction-based phrasing [5, 27]. We continue to pre-train this model using a mixture of language and motions data in both unsupervised and supervised manners: 1) To generalize

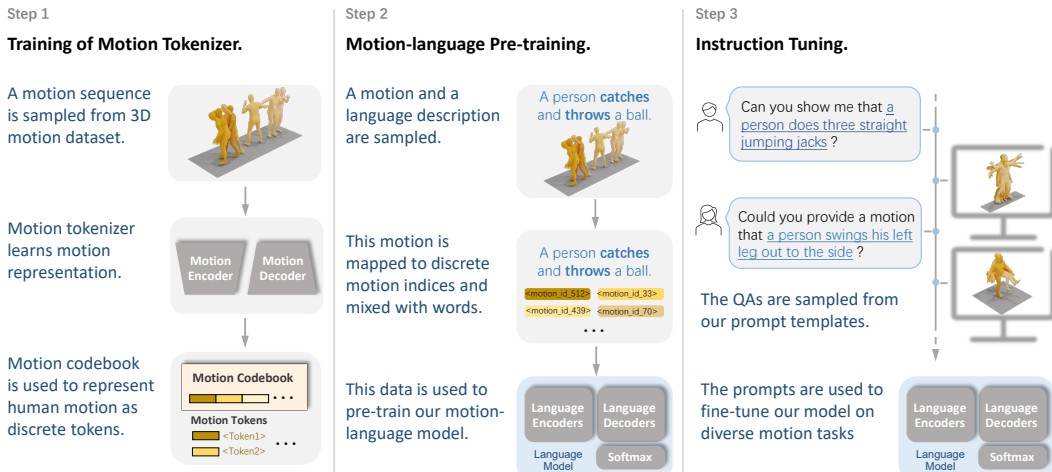

| Step 1 | Step 2 | Step 3 |
| --- | --- | --- |

**Training of Motion Tokenizer.** | **Motion-language Pre-training.** | **Instruction Tuning.**

Figure 3: Training Scheme. We introduce three training steps for our MotionGPT (Sec. 3.3): First $\mathcal{V}$ learn a codebook for discrete motion representation. Then we train language using a mixture of language and motion data to learn the semantic coupling between text and motion. Finally, we fine-tune the model in a multi-task text-motion dataset with instructions.

to various downstream tasks like [7, 35, 36, 27], we follow [36] to design an objective, where a certain percentage (15%) of tokens in the input tokens $X_s$ are randomly replaced with a special sentinel token. On the other side, the corresponding target sequence is constructed by extracting the dropped-out spans of tokens, delimited by the same sentinel tokens used in the input sequence, along with an additional sentinel token to indicate the end of the target sequence. 2) We then learn the motion-language relation by the supervision of paired text-motion datasets [10, 31]. We train MotionGPT on the supervised motion-language translation, where the input is either a human motion or a text description. After unsupervised and supervised training processes, we aim to equip our model with the understanding of text and motion relationships.

**Instruction Tuning Stage.** We construct a multi-task text-motion dataset by formulating it as instructions, building upon the foundation of existing text-to-motion datasets such as HumanML3D [10] and KIT [31]. Specifically, we define 15 core motion tasks, such as motion generation with text, motion captioning, motion prediction, and others. For each task, we compose dozens of different instruction templates, resulting in more than one thousand different tasks, each having a unique instruction prompt. For example, an instruction prompt for motion generation task could be "Can you generate a motion sequence that depicts 'a person emulates the motions of a waltz dance'?". Similarly, for the motion captioning task, the instruction prompt could be "Provide an accurate caption describing the motion of <motion_tokens>", where <motion_tokens> represents a sequence of motion tokens generated by our motion tokenizer. We have demonstrated the efficacy of instruction tuning in Sec. 4.3, which leads to improvement across various tasks and enhances the model performance for unseen tasks or prompts. More examples of prompts are provided in the supplements.

## 4 Experiments

Extensive comparisons evaluate the performance of our MotionGPTs across multiple motion-relevant tasks and datasets. Details of the dataset settings, evaluation metrics, and implementation specifics (Sec. 4.1) are provided. We first present a uniform benchmark by comparing our approach with other SOTAs across various tasks (Sec. 4.2). Then, we evaluate each specific comparison on text-to-motion (Sec. 4.2), motion-to-text (Sec. 4.2), motion prediction and motion in-between (Sec. 4.2). The supplements include more qualitative results, user studies, and further implementation details.

### 4.1 Experimental Setup

**Datasets.** General motion synthesis can support diverse task settings, and thus previous datasets and a modified benchmark are utilized to evaluate MotionGPT. The study primarily focuses on two text-to-motion datasets: HumanML3D [10] and KIT [31]. The KIT dataset provides 6,353 textual descriptions corresponding to 3,911 motion sequences, while the HumanML3D dataset [10] is a more recent dataset that contains 14,616 motion sequences obtained from AMASS [25], along with 44,970

| Methods | Text-to-Motion | | | Motion-to-Text | | | Motion Prediction | | Motion In-between | |
|---|---|---|---|---|---|---|---|---|---|---|
| | R TOP1↑ | FID↓ | DIV→ | R TOP3↑ | Bleu@4↑ | Cider↑ | FID↓ | DIV→ | FID↓ | DIV→ |
| Real | $0.511^{\pm.003}$ | $0.002^{\pm.000}$ | $9.503^{\pm.065}$ | 0.828 | - | - | 0.002 | 9.503 | 0.002 | 9.503 |
| MLD [52] | $0.481^{\pm.003}$ | $0.473^{\pm.013}$ | $9.724^{\pm.082}$ | - | - | - | - | - | - | - |
| T2M-GPT [57] | $\underline{0.491}^{\pm.003}$ | $\mathbf{0.116}^{\pm.004}$ | $9.761^{\pm.081}$ | - | - | - | 2.056 | 8.635 | - | - |
| TM2T [11] | $0.424^{\pm.017}$ | $1.501^{\pm.003}$ | $8.589^{\pm.076}$ | 0.823 | 7.00 | 16.8 | - | - | - | - |
| MDM [46] | $0.320^{\pm005}$ | $0.544^{\pm.044}$ | $\underline{9.559}^{\pm.086}$ | - | - | - | 6.031 | 7.813 | 2.698 | 8.420 |
| MotionGPT (Ours) | $\mathbf{0.492}^{\pm.003}$ | $\underline{0.232}^{\pm.008}$ | $\mathbf{9.528}^{\pm.071}$ | 0.827 | 12.47 | 29.2 | 0.905 | 8.972 | 0.214 | 9.560 |

Table 2: Comparison of four motion-related tasks on HumanML3D [10] dataset. The evaluation metrics are computed using the encoder introduced in [10]. The empty columns of previous methods indicate that they can not handle the task. The arrows ($\rightarrow$) indicate that closer to *Real* is desirable. **Bold** and underline indicate the best and the second best result on text-to-motion task.

sequence-level textual descriptions. To evaluate MotionGPT as a uniform framework on tasks, such as motion prediction and motion completion (in-between), we utilize the motion sequences available in HumanML3D, which is also a subset of the larger AMASS dataset. Following the previous works [10, 52, 46], we adopt the same motion representation for fair comparisons, which combines joint velocities, positions, and rotations. By using this consistent representation, MotionGPT enables the availability to support further studies in the field. ($cf.$ supplement for the benchmark details.)

**Evaluation Metrics** are summarized as four parts. (1) Motion quality: Frechet Inception Distance (FID) is our primary metric based on a feature extractor [10] to evaluate the distance of feature distributions between the generated and real motions. For motion completion, we utilize metrics used in motion prediction studies [56, 61, 24], such as Average Displacement Error (ADE) and Final Displacement Error (FDE), to evaluate the accuracy of the predicted motion. (2) Generation diversity: We utilize the Diversity (DIV) metric to assess the motions diversity, which calculates the variance through features extracted from the motions [10]. MultiModality (MM) measures the diversity of generated motions within the same text description of motion. (3) Text matching: Based on the feature space from [10], the motion-retrieval precision (R Precision) evaluates the accuracy of matching between texts and motions using Top 1/2/3 retrieval accuracy. Multi-modal Distance (MM Dist) measures the distance between motions and texts. (4) Linguistic quality: We follow [11] utilizing linguistic metrics from natural language studies, including BLUE [28], Rouge [23], Cider [49], and BertScore [60] to evaluate the quality of generated motion captions.

**Implementation Details.** We set the codebook of motion tokenizer as $K \in \mathbb{R}^{512 \times 512}$ for most comparisons. The motion encoder $\mathcal{E}$ incorporates a temporal downsampling rate $l$ of 4. We utilize T5 [36] as the underlying architecture for our language model, with a baseline model consisting of 12 layers in both the transformer encoder and decoder. The feed-forward networks have an output dimensionality of $d_{\text{ff}} = 3072$, and the attention mechanisms employ an inner dimensionality of $d_{\text{kv}} = 64$. The remaining sub-layers and embeddings have a dimensionality of $d_{\text{model}} = 768$. Moreover, all our models employ the AdamW optimizer for training. The motion tokenizers are trained utilizing a $10^{-4}$ learning rate and a 256 mini-batch size, while our language models have a $2 \times 10^{-4}$ learning rate for the pre-train stage, $10^{-4}$ for the instruction tuning stage, and a 16 mini-batch size for both stages. The motion tokenizer undergoes 150K iterations of training, while the language model undergoes 300K iterations during the pre-train stage and another 300K iterations during the instruction tuning stage. Small and Base models are trained on 8 Tesla V100 GPUs while Large models are trianed on 64 Tesla V100 GPUs.

### 4.2 Comparisons on Motion-relevant Tasks

**Comparisons on Multiple Tasks.** By introducing a uniform framework that treats human motion as a foreign language, we open up the exploration of diverse motion-relevant tasks. We employ a 220M pre-trained *Flan-T5-Base*[36, 5] model as our backbone and fine-tune the model through the pre-training and instruction tuning stage (Sec. 3.3) for all following comparisons. As shown in Tab. 2, we evaluate MotionGPT against state-of-the-art methods on key tasks such as text-conditioned motion generation [52, 57, 11, 46], motion captioning [11], motion prediction [46], and motion in-between[46]. While we leverage existing results from previous works or benchmarks for text-to-motion and motion-to-text tasks, we re-implement the motion diffusion models [46] for motion prediction and evaluate it under the same metrics and settings. Please note that some methods are

| Methods | RPrecision↑ | | | FID↓ | MMDist↓ | Diversity→ | MModality↑ |
|---|---|---|---|---|---|---|---|
| | Top1 | Top2 | Top3 | | | | |
| Real | $0.511^{\pm.003}$ | $0.703^{\pm.003}$ | $0.797^{\pm.002}$ | $0.002^{\pm.000}$ | $2.974^{\pm.008}$ | $9.503^{\pm.065}$ | - |
| TM2T [11] | $0.424^{\pm.003}$ | $0.618^{\pm.003}$ | $0.729^{\pm.002}$ | $1.501^{\pm.017}$ | $3.467^{\pm.011}$ | $8.589^{\pm.076}$ | $2.424^{\pm.093}$ |
| T2M [10] | $0.457^{\pm.002}$ | $0.639^{\pm.003}$ | $0.740^{\pm.003}$ | $1.067^{\pm.002}$ | $3.340^{\pm.008}$ | $9.188^{\pm.002}$ | $2.090^{\pm.083}$ |
| MotionDiffuse [58] | $\underline{0.491}^{\pm.001}$ | $\mathbf{0.681}^{\pm.001}$ | $\mathbf{0.782}^{\pm.001}$ | $0.630^{\pm.001}$ | $\underline{3.113}^{\pm.001}$ | $9.410^{\pm.049}$ | $1.553^{\pm.042}$ |
| MDM [46] | $0.320^{\pm.005}$ | $0.498^{\pm.004}$ | $0.611^{\pm.007}$ | $0.544^{\pm.044}$ | $5.566^{\pm.027}$ | $\underline{9.559}^{\pm.086}$ | $\underline{2.799}^{\pm.072}$ |
| MLD [52] | $0.481^{\pm.003}$ | $0.673^{\pm.003}$ | $0.772^{\pm.002}$ | $0.473^{\pm.013}$ | $3.196^{\pm.010}$ | $9.724^{\pm.082}$ | $2.413^{\pm.079}$ |
| T2M-GPT [57] | $\underline{0.491}^{\pm.003}$ | $0.680^{\pm.003}$ | $0.775^{\pm.002}$ | $\mathbf{0.116}^{\pm.004}$ | $3.118^{\pm.011}$ | $9.761^{\pm.081}$ | $1.856^{\pm.011}$ |
| MotionGPT (Pre-trained) | $0.435^{\pm.003}$ | $0.607^{\pm.002}$ | $0.700^{\pm.002}$ | $\underline{0.160}^{\pm.008}$ | $3.700^{\pm.009}$ | $9.411^{\pm.081}$ | $\mathbf{3.437}^{\pm.091}$ |
| MotionGPT (Fine-tuned) | $\mathbf{0.492}^{\pm.003}$ | $\mathbf{0.681}^{\pm.003}$ | $\underline{0.778}^{\pm.002}$ | $0.232^{\pm.008}$ | $\mathbf{3.096}^{\pm.008}$ | $\mathbf{9.528}^{\pm.071}$ | $2.008^{\pm.084}$ |

Table 3: Comparison of text-to-motion on HumanML3D [10]. The empty MModality indicates *Real* motion is deterministic. These methods are sorted by FID. *Pre-trained* and *Fine-tuned* indicate uniform motion-language pre-training and specific fine-tuning on this task. ($cf$. Tab. 2 for notations.)

| Methods | RPrecision↑ | | MMDist↓ | $\text{Length}_{avg}$↑ | Bleu@1↑ | Bleu@4↑ | Rouge↑ | Cider↑ | BertScore↑ |
|---|---|---|---|---|---|---|---|---|---|
| | Top1 | Top3 | | | | | | | |
| Real | 0.523 | 0.828 | 2.901 | 12.75 | - | - | - | - | - |
| TM2T[11] | 0.516 | 0.823 | 2.935 | 10.67 | **48.9** | 7.00 | **38.1** | 16.8 | 32.2 |
| MotionGPT (Ours) | **0.543** | **0.827** | **2.821** | **13.04** | 48.2 | **12.47** | 37.4 | **29.2** | **32.4** |

Table 4: Comparison of motion captioning on HumanML3D [10]. The evaluation metrics follow [11], while we use the ground truth texts without pre-processing for linguistic metrics calculation.

designed for specific tasks, and thus some metrics are empty for tasks they cannot handle. The results presented in Tab. 2 demonstrate that our MotionGPT achieves competitive performance across all evaluated tasks, highlighting its capability to address diverse motion tasks within a single model.

**Comparisons on Text-to-Motion.** The text-to-motion task involves generating human motion sequences based on a given text input. We evaluate the proposed the MotionGPT model as the pre-trained MotionGPT, the same one in Tab. 2, as well as fine-tuned it on text-to-motion task. We compare our MotionGPTs with other SOTAs [11, 10, 46, 52, 57] and evaluate the performance on both HumanML3D and KIT datasets using suggested metrics [10]. The results are computed with a 95% confidence interval, obtained from 20 repeated runs. The majority of the reported results are taken directly from their own papers or the benchmark presented in [10]. Tab. 3 summarizes the comparison results, where MotionGPT achieves competitive performance on most metrics.

**Comparisons on Motion-to-Text.** The motion-to-text task involves generating a text description based on a given human motion sequence. We compare the pre-trained MotionGPT with recent work TM2T [11]. We evaluate the performance on the HumanML3D using the suggested metrics from [11]. Additionally, we measure the average numbers of words $\text{Length}_{avg}$ for further comparisons. Please note that the reported results in [11] are evaluated with pre-processed ground truth text, which ignores the grammatical tense and plural forms of words. In Tab. 4, we directly use the ground truth text descriptions for a more accurate assessment. This comparison shows that MotionGPT overperforms recent work on text descriptions of given motions.

**Comparisons on Motion Prediction and In-between.** We summarize motion prediction and in-between together as general motion completion. To evaluate the motion completion capability of MotionGPT, we employ part of the AMASS dataset [25], a motion-only dataset. For motion prediction task, we only input around the first 20% of the motion sequence as conditions. For in-between, we mask about 50% motion randomly for completion. We also fine-tune MotionGPT specifically for this task and employ FID, ADE, and FDE as metrics like Sec. 4.1. Furthermore, we evaluate MDM [46] on motion prediction by utilizing their provided model, which also supports motion in-between through masked motion "in-painting". The real motion data is used as one of our baselines. Tab. 5 reports that our MotionGPT has the best motion completion quality and diversity.

| Methods | Motion Prediction | | | | Motion In-between | | |
|---|---|---|---|---|---|---|---|
| | FID ↓ | Diversity↑ | ADE↓ | FDE↓ | FID ↓ | Diversity↑ | ADE↓ |
| Real | 0.002 | 9.503 | - | - | 0.002 | 9.503 | - |
| MDM[46] | 6.031 | 7.813 | 5.446 | 8.561 | 2.698 | 8.420 | 3.787 |
| T2M-GPT[57] | 2.056 | 8.635 | 6.161 | 8.302 | - | - | - |
| MotionGPT (Ours) | **0.905** | **8.972** | **4.745** | **6.040** | **0.214** | **9.560** | **3.762** |

Table 5: Comparison of motion prediction and motion in-between on part of AMASSS [25] dataset using motion data only. FID indicates motion quality and Diversity (DIV) for motion diversity within each condition. ADE and FDE are joints distance between generation and ground truth.

| Size | Instruction Tuning | Text-to-Motion | | | Motion-to-Text | | | Motion Prediction | | Motion In-between | |
|---|---|---|---|---|---|---|---|---|---|---|---|
| | | R TOP3 ↑ | FID ↓ | DIV → | MMDist↓ | Bleu@4↑ | Cider↑ | FID ↓ | DIV → | FID ↓ | DIV → |
| Real | - | 0.797 | 0.002 | 9.503 | 2.901 | - | - | 0.002 | 9.503 | 0.002 | 9.503 |
| Small | | 0.706 | 0.727 | 9.264 | **2.748** | 12.02 | 24.9 | - | - | - | - |
| Small | ✓ | 0.663 | 0.336 | 9.239 | 2.931 | 10.54 | 24.3 | 0.954 | 8.727 | 0.326 | 9.618 |
| Base | | **0.722** | 0.365 | 9.407 | 2.821 | **12.47** | **29.2** | - | - | - | - |
| Base | ✓ | 0.700 | 0.160 | **9.411** | 3.019 | 11.42 | 28.2 | 0.905 | 8.972 | **0.214** | **9.560** |
| Large | | 0.694 | 0.234 | 9.310 | 2.776 | 12.44 | 28.5 | - | - | - | - |
| Large | ✓ | 0.708 | **0.159** | 9.301 | 3.011 | 11.71 | 29.1 | **0.556** | **8.975** | 0.223 | 9.358 |

Table 6: Evaluation of instruction tuning and different model sizes of MotionGPTs in four motion tasks on HumanML3D [10] dataset. ($cf$. Tab. 2 for metrics details)

## 4.3 Ablation Studies

MotionGPT employs T5 [36] as the motion-aware language backbone model, and we train these models with pre-training and then instruction tuning. Thus, both model size and training strategy influence the performance of MotionGPTs. We here evaluate them on the typical motion tasks. More detailed ablation studies are provided in the supplements.

**Model Sizes.** We evaluate the performance of models with different sizes across four motion tasks. Besides the base 220M MotionGPT in Sec. 4.1, we now evaluate 60M, 220M, and 770M MotionGPTs. Tab. 6 demonstrates that the 220M base model has achieved remarkable performance compared to the smaller 60M model. However, the larger model size of current Motions does not yield significant improvements and, in few cases, even leads to worse results, as observed in the motion in-between task. We believe this could be caused by the small amount of current motion datasets. HumanML3D only includes 15k motion sequences, much smaller than even billions of language and image data.

**Effectiveness of Instruction Tuning.** We evaluate the impact of our instruction tuning strategy on different model sizes. The results in Tab. 6 demonstrate that instruction tuning enhances the versatility of MotionGPT, enabling more motion tasks like motion completion and improving the motion performance of the text-to-motion task. However, for pure text-generation tasks, the model performance is downgraded, likely due to the pair amount of textual descriptions and coupled motions.

## 5 Disscusion

As the first trial, to our best knowledge, exploring human motion generation using language models, the proposed MotionGPT still owns limitations as follows. MotionGPT only utilizes motion on articulated human bodies, while many other works focus on faces [16, 4], hands [39, 22, 21] and even animal [40, 62] motion. Besides, our method is also restricted to multiple humans without modeling human-object, or human-environment interactions [41]. It is interesting to model the human interaction scenarios in a motion-language framework and generate controllable motions [41].

We summarize the proposed MotionGPT as a uniform motion-language framework to generate plausible human motion and natural language descriptions through prompt-based instructions. Compared to the compatible motion diffusion methods [52, 46], our MotionGPT produces competitive results on motion generation, motion captioning, motion prediction, and motion in-between using only one pre-trained generative model. With the advancement of large language data and models [36, 5], MotionGPT is also capable of addressing natural question-to-answer tasks. Extensive experiments on various human motion-relevant tasks demonstrate the effectiveness and extendibility of MotionGPT.

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
