# Appendix of MotionGPT

This appendix provides qualitative comparison results (Sec. A), additional experiments (Sec. B) on the components of MotionGPT models, inference time (Sec. C), statistics on motion vocabulary (Sec. D), evaluations on hyperparameters (Sec. E), user study (Sec. F), a protocol for the uniform evaluation (Sec. G), and more implementation details (Sec. H) of MotionGPT models . Please note evaluations on our training scheme (Sec. B.2), elaborations on the difference of T2M-GPT (Sec. B.6), implementation details of motion completion (Sec. B.7), and more metric definitions (Sec. G).

**Video.** We have provided supplemental videos on Project Page. In these supplemental videos, we show 1) comparisons of text-to-motion, 2) comparisons of motion captioning, and 3) more results on motion prediction and other tasks. We suggest watching this video for dynamic motion results.

**Code** is available on GitHub Repo. We provide the process of the training and evaluation of MotionGPT models, the pre-trained models and the demo scripts. The live demo is on Huggingface.

## A   Qualitative Results

We visualize some qualitative results on the comparison of text-to-motion ($cf$. Fig. 4), motion-to-text ($cf$. Fig. 6), and our result gallery on multiple tasks ($cf$. Fig. 5).

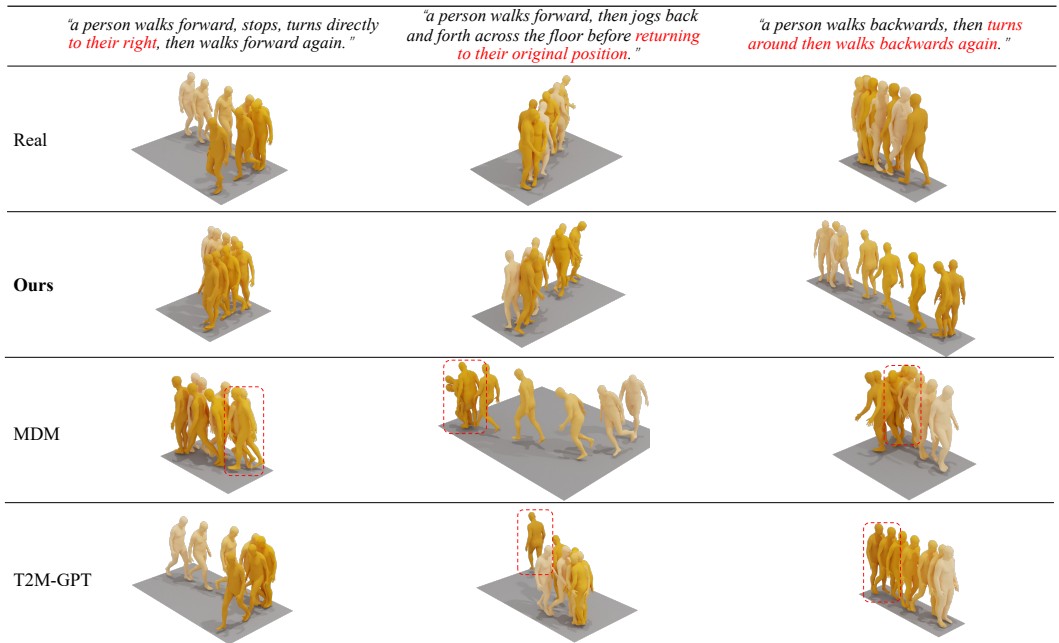

Figure 4: Comparison on text-driven motion generation. The provided state-of-the-art methods are under the same training and inference setting on HumanML3D [2]. The red words and boxes highlight the misaligned motions. The results demonstrate that our motion-language per-training shows promising text understanding for motion generation.

37th Conference on Neural Information Processing Systems (NeurIPS 2023).

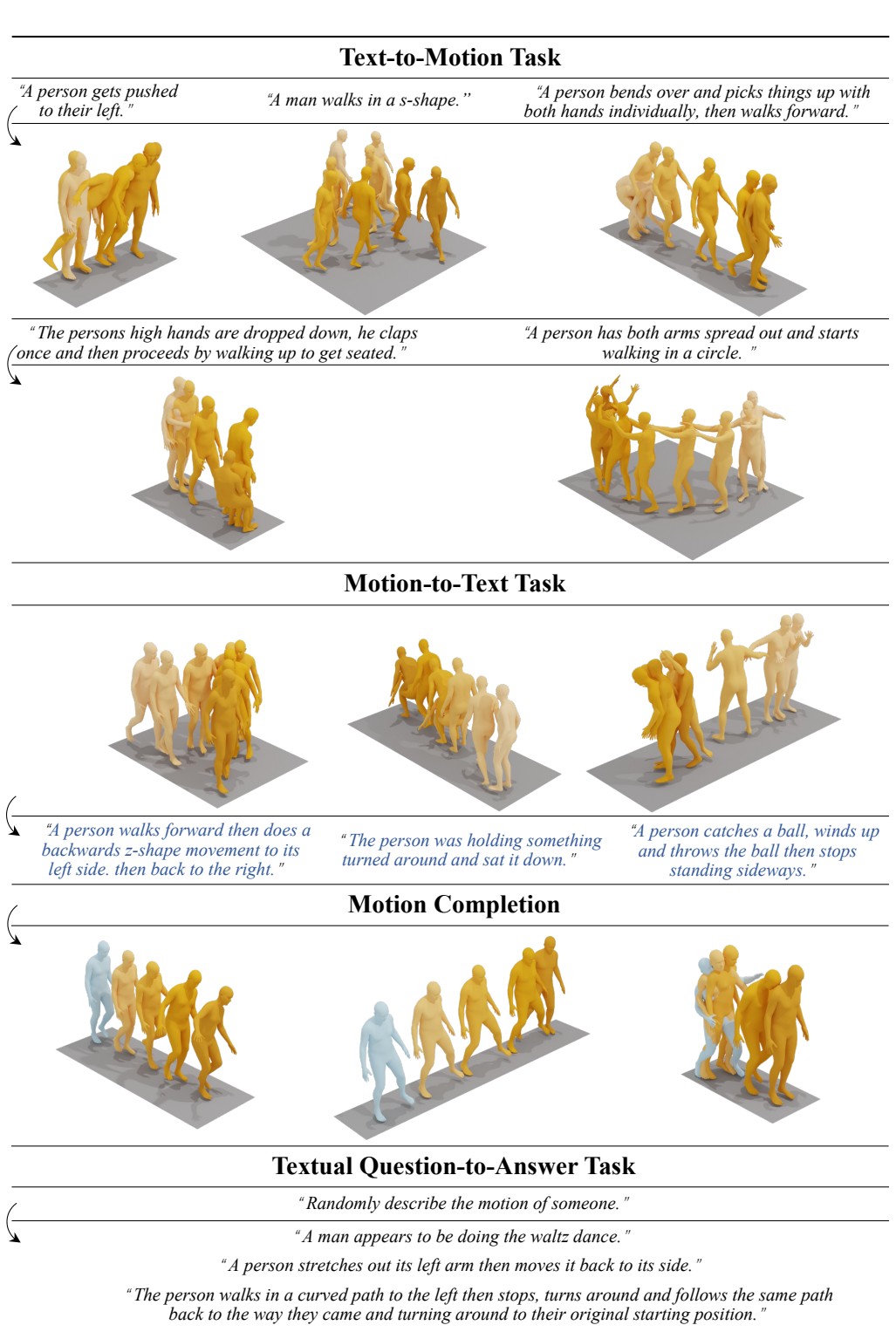

Figure 5: Gallery for the results of our unified MotionGPT. More samples are from our best model for text-to-motion synthesis, motion captioning, and textual question-to-answer task. The supervision of MotionGPT relies on our instruction-based motion-language dataset ($cf$. Sec. G) based on previous motion datasets [2, 6]. We recommend the dynamic visualization in our supplemental video.

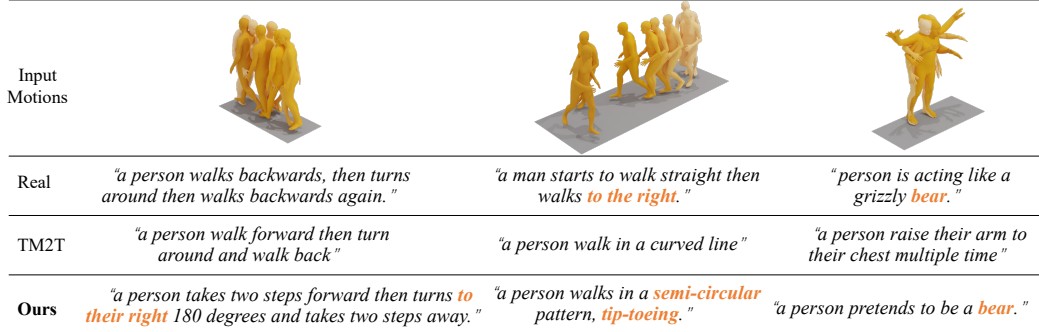

| | Input Motions | | |
|---|---|---|---|
| Real | *"a person walks backwards, then turns around then walks backwards again."* | *"a man starts to walk straight then walks **to the right**."* | *"person is acting like a grizzly **bear**."* |
| TM2T | *"a person walk forward then turn around and walk back"* | *"a person walk in a curved line"* | *"a person raise their arm to their chest multiple time"* |
| **Ours** | *"a person takes two steps forward then turns **to their right** 180 degrees and takes two steps away."* | *"a person walks in a **semi-circular** pattern, **tip-toeing**."* | *"a person pretends to be a **bear**."* |

Figure 6: Comparison of the state-of-the-art method on motion captioning task. All provided methods are under the same training and inference setting on HumanML3D [2]. The results demonstrate that our text descriptions correspond better to the motion and have correct grammar. The orange words indicate the matching results, while the red marks the incorrect grammar.

## B    Additional Experiments

We conduct several experiments to continue the evaluations of MotionGPT models. We first evaluate the text-to-motion results on KIT dataset (Sec. B.1). Then we evaluate the hyperparameters of motion tokenizer $\mathcal{V}$ (Sec. B.2). After that, we study the effectiveness of the training scheme (Sec. B.2). We also provide the elaboration on the difference of T2M-GPT (Sec. B.6), implementation details of motion completion (Sec. B.7).

### B.1    Text-to-Motion on KIT dataset.

Following the same procedure on HumanML3D[2] dataset, We train a 220M MotionGPT base model on the KIT[10] dataset without any pre-training. We evaluate this model under the same settings of [2]. Most results are borrowed from their own paper of the benchmark in [2]. Tab. 7 shows that MotionGPT achieves comparable performance compared to the previous state-of-the-arts.

| Methods | RPrecision↑ | | | FID↓ | MMDist↓ | Diversity→ | MModality↑ |
|---|---|---|---|---|---|---|---|
| | Top1 | Top2 | Top3 | | | | |
| Real | $0.424^{\pm.005}$ | $0.649^{\pm.006}$ | $0.779^{\pm.006}$ | $0.031^{\pm.004}$ | $2.788^{\pm.012}$ | $11.08^{\pm.097}$ | - |
| TM2T[3] | $0.280^{\pm.005}$ | $0.463^{\pm.006}$ | $0.587^{\pm.005}$ | $3.599^{\pm.153}$ | $4.591^{\pm.026}$ | $9.473^{\pm.117}$ | $3.292^{\pm.081}$ |
| MDM[14] | $0.164^{\pm.004}$ | $0.291^{\pm.004}$ | $0.396^{\pm.004}$ | $0.497^{\pm.021}$ | $9.191^{\pm.022}$ | $10.85^{\pm.109}$ | $1.907^{\pm.214}$ |
| MLD[17] | $0.390^{\pm.008}$ | $0.609^{\pm.008}$ | $0.734^{\pm.007}$ | $0.404^{\pm.027}$ | $3.204^{\pm.027}$ | $10.80^{\pm.117}$ | $2.192^{\pm.071}$ |
| T2M-GPT [19] | $0.416^{\pm.006}$ | $0.627^{\pm.006}$ | $0.745^{\pm.006}$ | $0.514^{\pm.029}$ | $3.007^{\pm.023}$ | $10.92^{\pm.108}$ | $1.570^{\pm.039}$ |
| MotionGPT (Ours) | $0.366^{\pm.005}$ | $0.558^{\pm.004}$ | $0.680^{\pm.005}$ | $0.510^{\pm.016}$ | $3.527^{\pm.021}$ | $10.35^{\pm.084}$ | $2.328^{\pm.117}$ |

Table 7: We involve KIT [10]dataset and evaluate the methods on the text-driven motion generation task. Please refer to Tab. 3 for more details on metrics and notations.

### B.2    Ablation on Motion Tokenizer.

We ablate the motion tokenizer $\mathcal{V}$ of our MotionGPT models, studying the size $K$ of motion codebooks. We also compare this VQ-VAE with other VAE models in previous works [8, 9, 17], as shown in Tab. 13. This comparison demonstrates the improvement of VQ-VAE on motion reconstruction. With this ablation studies on the codebook size $K$, we thus select $K = 512$ for most experiments.

| Method | Reconstruction | | | | |
|---|---|---|---|---|---|
| | MPJPE↓ | PAMPJPE↓ | ACCL↓ | FID↓ | DIV→ |
| Real | - | - | - | 0.002 | 9.503 |
| VPoser-t [8] | 75.6 | 48.6 | 9.3 | 1.430 | 8.336 |
| ACTOR [9] | 65.3 | 41.0 | **7.0** | 0.341 | **9.569** |
| MLD-1 [17] | **54.4** | 41.6 | 8.3 | 0.247 | 9.630 |
| MotionGPT (Ours) | 55.8 | **40.1** | 7.5 | **0.067** | 9.675 |
| $K = 256$ | 76.4 | 51.3 | 10.0 | 0.187 | **9.496** |
| $K = 512$ | **55.8** | **40.1** | **7.5** | **0.067** | 9.675 |
| $K = 1024$ | 60.3 | 44.0 | 8.6 | 0.086 | 9.677 |
| $K = 2048$ | 78.9 | 51.4 | 10.5 | 0.141 | 9.597 |

Table 8: Evaluation of our motion tokenizer on the motion part of HumanML3D [2] dataset. We follow MLD [17] to evaluate our VQ-VAE model $\mathcal{V}$: MPJPE and PAMPJPE are measured in millimeter. ACCL indicates acceleration error. We evaluate FID and Diversity the same as Tab. 3. The baselines of VPoser-t [8] and ACTOR [9] are borrowed from MLD. $K$ indicates the codebook size, and $K = 512$ shows the best performance of motion reconstruction.

## B.3 Effectiveness of Training Scheme

**Motion-Language Pre-training** vs **Instructions Tuning**. We have provided the illustration of our training scheme in Fig. 3 and the evaluation in Tab. 6. We further ablate this training scheme on the base MotionGPT model, by evaluating the motion-language pre-training (the second step) and instruction tuning (the third step). As shown in Tab. 9, we train these models with the same 600K iterations. Compared to other training combinations, the full-stage MotionGPT achieves higher performance on most motion tasks.

| Size | Pre-training | Instruction Tuning | Text-to-Motion | | | Motion-to-Text | | | Motion Prediction | | Motion In-between | |
|---|---|---|---|---|---|---|---|---|---|---|---|---|
| | | | R TOP3 ↑ | FID ↓ | DIV → | MMDist↓ | Bleu@4↑ | Cider↑ | FID ↓ | DIV → | FID ↓ | DIV → |
| Real | - | - | 0.797 | 0.002 | 9.503 | 2.901 | - | - | 0.002 | 9.503 | 0.002 | 9.503 |
| Base | ✔ | ✗ | **0.722** | 0.365 | 9.407 | **2.821** | **12.47** | **29.2** | - | - | - | - |
| Base | ✔ | ✔ | 0.700 | **0.160** | **9.411** | 3.019 | 11.42 | 28.2 | 0.905 | 8.972 | **0.214** | **9.560** |
| Base | ✗ | ✔ | 0.607 | 0.324 | 9.563 | 3.374 | 10.92 | 27.7 | 1.643 | 8.829 | 0.323 | 9.628 |

Table 9: Evaluation of the training scheme on the base MotionGPT models. We evaluate the results with the proposed evaluation protocols in Sec. G. Please refer to Tab. 2 for metrics and the details.

**Instructions Tuning** vs **Task-Specific Tuning**. While our unified instruction-tuned MotionGPT model has demonstrated competitive performance across various motion-related tasks, further fine-tuning can always enhance its performance on specific tasks. Therefore, we focus on the text-to-motion task and motion in-between task as illustrative examples to showcase the performance of the model before and after fine-tuning. By comparing the results in Tab. 10, we can assess the effectiveness of fine-tuning in improving task-specific performance.

| Insturct tuned | Fine tuned | Text-to-Motion | | | Motion In-between | | |
|---|---|---|---|---|---|---|---|
| | | R TOP1↑ | FID↓ | DIV→ | FID ↓ | DIV↑ | ADE↓ |
| ✔ | ✗ | 0.435 | **0.160** | 9.411 | 0.214 | **9.560** | 3.762 |
| ✔ | ✔ | **0.492** | 0.232 | **9.528** | **0.209** | 9.378 | **3.281** |

Table 10: Evaluation of new task tuning of different size models on HumanML3D [2] dataset.

## B.4 Different Backbone of Language Model

The first language model that we used to build MotionGPTs is LLaMA-13B [15]. However, it shows insufficient performance and low training efficiency. We assume the reason is the limited dataset size compared to the large parameters and language data of LLaMA. We tried a smaller size decoder-only backbone GPT2-Medium [12] and provide the results in Tab. 11. Then, we thus chose T5-770M, a small but common language model, as our final backbone, because many previous vision-language multimodal works, like Unified-IO and BLIP, have chosen T5, this encoder-decoder architecture. It shows a strong power to address multi-modal tasks. In addition, the decoder-only model has the advantage for self-supervised without pair data while we have paired data which this advance is greatly weakened. We are still working on collecting a large motion dataset for larger motion-language models.

| Backbone | Type | Parameters | Text-to-Motion | | | Motion-to-Text | | | Motion Prediction | | Motion In-between | |
|---|---|---|---|---|---|---|---|---|---|---|---|---|
| | | | R TOP3 ↑ | FID ↓ | DIV → | MMDist↓ | Bleu@4↑ | Cider↑ | FID ↓ | DIV → | FID ↓ | DIV → |
| Real | - | - | 0.797 | 0.002 | 9.503 | 2.901 | - | - | 0.002 | 9.503 | 0.002 | 9.503 |
| T5-Base | Encoder-Decoder | 220M | 0.700 | 0.160 | **9.411** | 3.019 | 11.42 | 28.2 | 0.905 | 8.972 | **0.214** | **9.560** |
| T5-Large | Encoder-Decoder | 770M | **0.708** | **0.159** | 9.301 | 3.011 | 11.71 | 29.1 | **0.556** | **8.975** | 0.223 | 9.358 |
| GPT2-Medium | Decoder-only | 355M | 0.508 | 0.258 | 9.274 | 4.923 | 7.44 | 17.3 | 0.794 | 8.692 | 0.241 | 9.185 |

Table 11: Evaluation of different backbone of MotionGPTs on HumanML3D [11] dataset. Please refer to Tab. 2 for metrics and the details.

## B.5 Zero-shot and Failure Cases

Unlike the previous motion generators using the text encoder of CLIP [11] for conditions, please note that MotionGPTs leverage language models to learn the motion-language relationship, instead of relying on text features from CLIP. According to our zero-shot results ($cf$. Fig. 7) and performances on multi-tasks ($cf$. Tab. 2), MotionGPTs establish robust connections between simple/complex texts and simple motions in evaluations, but they fall short when it comes to complex-text to complex motion translation.

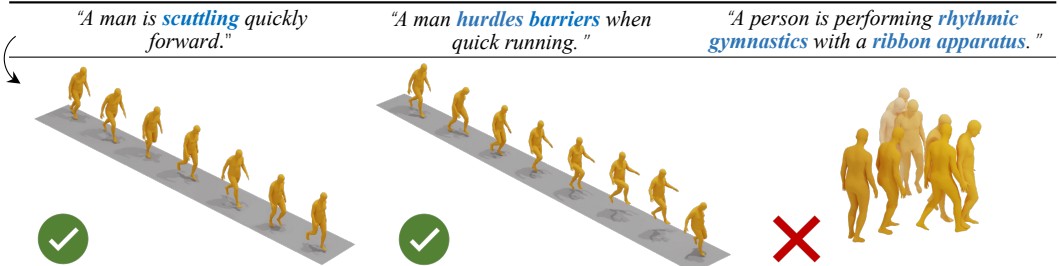

Figure 7: We provide zero-shot cases and failure cases. Benefitting from strong language models, MotionGPTs can understand unseen works in the text-to-motion training set, like "scuttling" and "barriers", and generate correct motions based on the meaning of sentences. However, it still struggles to generate unseen motions, like gymnastics, even if MotionGPTs understand the text inputs.

## B.6 Difference between T2M-GPT

We introduce the difference between T2M-GPT [19] to show our unified framework. T2M-GPT investigates a generative framework based on VQ-VAE and Transformer for motion generation only. They incorporate language information by leveraging CLIP [11] to extract text embedding as motion generation conditions, which is similar to most previous work, such as MDM [14], MLD [17], and MotionDiffuse [20]. However, our MotionGPTs are based on the pre-trained language model so it naturally leverages the strong language generation and zero-shot transfer abilities of pre-trained language models. Benefiting from the motion-language vocabulary, MotionGPT thus generates both human language and human motion in a unified model.

### B.7 Implementation details of Motion Completion

Please note that MDM[14] accomplish motion in-between task in their paper through masked motion "in-painting" which fix the first and last 25% of the motion, leaving the model to generate the remaining 50% in the middle. To achieve the motion prediction task with MDM, we fix the first 20% of the motion and then generate the remaining. All our results are computed by utilizing their provided pre-trained model. To compare with MDM in Tab. 5 on both motion in-between and motion prediction tasks, we evaluate our MotionGPT with the same setting during the inference.

## C    Inference Time

We provide a detailed study on inference time with our different model sizes below. Due to our auto-regressive model for motion generation, we use Frames Per Second (FPS) to evaluate our time costs. All the time costs are evaluated on 8 Tesla V100 using one batch size. Tab. 12 shows that any size of our MotionGPTs can support real-time human animations and come up to hundreds of FPS.

| Models | Backbone | Parameters | FPS ↑ |
|---|---|---|---|
| MotionGPT | Small | 60 M | 421.31 |
| MotionGPT | Base | 220 M | 222.69 |
| MotionGPT | Large | 770 M | 119.75 |

Table 12: Evaluation of inference time costs on text-driven motion generation. We evaluate the Frames Per Second (FPS) by averaging our generated frames for each second. We show the time costs on different model sizes. Under the same 1 Tesla V100, the smaller model size gets the faster FPS. All models can support real-time motion animation applications.

## D    Motion Vocabulary

### D.1    Statistics on Motion Vocabulary

We visualize the usage of each "word" in our motion vocabulary $V_m$ item generated by our motion tokenizer $\mathcal{V}$. We sample all motions from the whole test set of HumanML3D dataset [2] and count each "word". In Fig. 8, it shows the utilization of our motion codebook, which seems to be a concise but informative motion vocabulary.

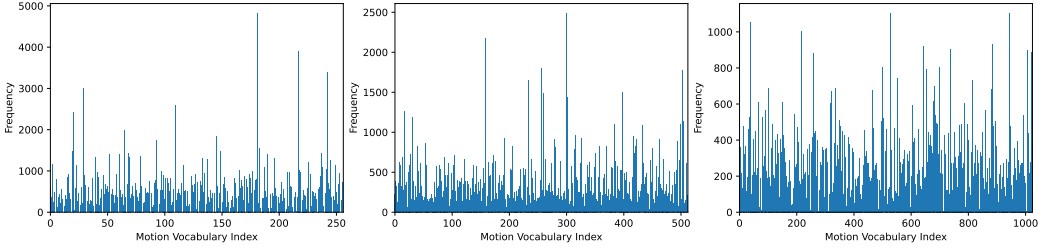

Figure 8: The statistics of each "word" in different sizes of motion vocabulary $V_m$. From left to right, the vocabulary size is $K = 256, 512, 1024$. ($cf$. Tab. 13, $K = 512$ for the best motion quality.)

### D.2    Visualization of Motion Vocabulary

As shown in  Fig. 9, we visualize these motion tokens in motion vocabulary and their corresponding localized spatial-temporal contexts, depicted within 4-frame motion segments.

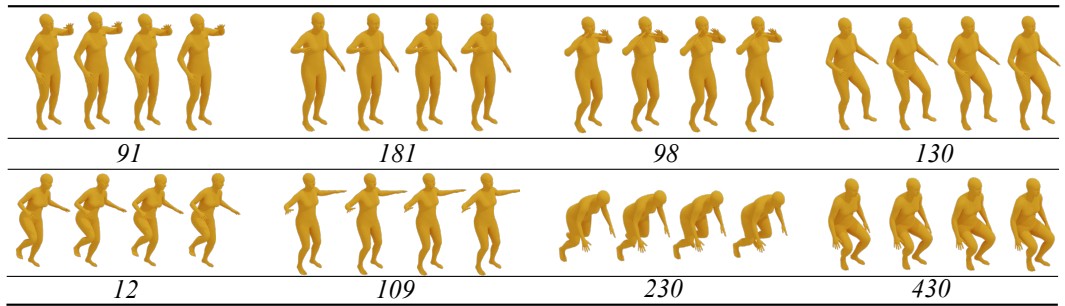

| | 91 | 181 | 98 | 130 |
| | 12 | 109 | 230 | 430 |

Figure 9: We visualize these motion tokens in motion vocabulary $V_m$ and their corresponding localized spatial-temporal contexts, depicted within 4-frame motion segments.

| Downsampling | Reconstruction | | | | |
|---|---|---|---|---|---|
| | MPJPE↓ | PAMPJPE↓ | ACCL↓ | FID↓ | DIV→ |
| $l = 1$ | 76.2 | 49.5 | 19.5 | 0.421 | 9.613 |
| $l = 2$ | **52.6** | **37.7** | **9.5** | 0.135 | 9.722 |
| $l = 4$ | 55.8 | 40.1 | 7.5 | **0.067** | 9.675 |
| $l = 8$ | 62.7 | 45.3 | 8.7 | 0.223 | **9.584** |

Table 13: Evaluation of our motion tokenizer on the motion part of HumanML3D [2] dataset. We follow MLD [17] to evaluate our VQ-VAE model $\mathcal{V}$: MPJPE and PAMPJPE are measured in millimeter. ACCL indicates acceleration error. We evaluate FID and Diversity the same as Tab. 3. The baselines of VPoser-t [8] and ACTOR [9] are borrowed from MLD. $K$ indicates the codebook size, and $K = 512$ shows the best performance of motion reconstruction.

# E   Evaluation of Hyperparameters

## E.1   Evaluation of Different Down-sample Rate

We selected the down-sample rate based on the frames-per-second (FPS) of the HumanML3D and KIT-ML datasets, which is 20 fps. Therefore, down-sampling by a factor of 4 to achieve 5 fps can ensure distinctiveness in motion frames, and prevents redundancy, and acceleration training. This choice was also made to ensure a fair comparison, as we utilized the same down-sample rate as T2M-GPT. As shown in , we provide an ablation study on these parameters, where a factor of 4 achieves the best Frechet Inception Distance (FID) in motion reconstructions.

## E.2   Evaluation of Different Sampling Strategies

We conduct experiments to investigate the impact of different sampling strategies on the generation results. Specifically, we compare the use of greedy search, which selects the most probable token at each step, with sampling from the probability distribution and adopting beam search, which is evaluated in previous language models [13]. Beam search expands the search space for improved sequence probability matching. The results in Tab. 14 demonstrate that while avoiding sampling and using beam search can slightly improve generation quality, they also significantly reduce the diversity of generated motions from the same text description.

| Method | Sample | #beams | R Precision Top 3↑ | FID↓ | MM Dist↓ | Diversity→ | MModality↑ |
|---|---|---|---|---|---|---|---|
| Real | - | - | $0.797^{\pm.002}$ | $0.002^{\pm.000}$ | $2.974^{\pm.008}$ | $9.503^{\pm.065}$ | - |
| MotionGPT | | - | $0.780^{\pm.002}$ | $0.224^{\pm.009}$ | $3.076^{\pm.009}$ | $9.492^{\pm.056}$ | - |
| | | 2 | $0.780^{\pm.002}$ | $0.199^{\pm.008}$ | $3.083^{\pm.007}$ | $9.512^{\pm.063}$ | - |
| | | 3 | $0.781^{\pm.002}$ | $0.179^{\pm.008}$ | $3.099^{\pm.009}$ | $9.516^{\pm.064}$ | - |
| | | 4 | $0.782^{\pm.002}$ | $0.160^{\pm.007}$ | $3.092^{\pm.010}$ | $9.536^{\pm.060}$ | - |
| MotionGPT | ✓ | - | $0.778^{\pm.002}$ | $0.232^{\pm.008}$ | $3.096^{\pm.008}$ | $9.528^{\pm.071}$ | $2.008^{\pm.084}$ |
| | ✓ | 2 | $0.780^{\pm.002}$ | $0.194^{\pm.008}$ | $3.091^{\pm.010}$ | $9.508^{\pm.063}$ | $1.140^{\pm.064}$ |
| | ✓ | 3 | $0.780^{\pm.002}$ | $0.190^{\pm.008}$ | $3.089^{\pm.011}$ | $9.529^{\pm.061}$ | $0.929^{\pm.055}$ |
| | ✓ | 4 | $0.780^{\pm.002}$ | $0.182^{\pm.008}$ | $3.093^{\pm.008}$ | $9.537^{\pm.059}$ | $0.803^{\pm.044}$ |

Table 14: Evaluations on hyperparameters for MotionGPT generations. We study the influence of two hyperparameters: *sample* stands for sampling from distribution; *#beams* means the number of beams for beam search, where empty means no beam search.

# F  User Study

We achieve a detailed user study to evaluate our model's performance. For text-to-motion assessment, we generated motions for 80 HumanML3D [2] test set descriptions, comparing MotionGPTs with MDM [14] and T2M-GPT [19], alongside GT. Semantic and realism studies presented text-video pairs to participants, asking which motion corresponded better or was more realistic, respectively. In the motion-to-text study, we visualized 50 GT motions with GT descriptions and generated corresponding textual descriptions using TM2T [3] and our method. Each participant addressed a batch of questions randomly from all questions, and 19 unqualified participants among a total of 110 samples were identified and excluded by 2 'catch trials' questions. Each video pairs were reviewed by multiple participants, with a majority vote determining superior methods. Equal scores were assigned for tied results. As shown in Fig. 10, in both two tasks, our MotionGPT was preferred over the other state-of-the-art methods and even competitive with the ground truth.

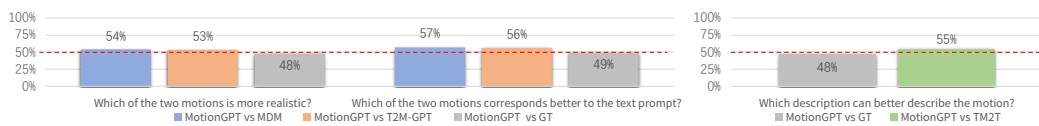

Figure 10: User Study. We investigate our motion quality and the alignment with test descriptions. The left part is the user study for text-to-motion. The right part is for motion captioning.

# G  Evaluation Protocols on the Uniform Motion-Language Generation.

We propose a protocol to evaluate our unified MotionGPT on multiple motion-language generation tasks. Upon previous datasets [2, 10, 6], we build an instruction motion-language dataset, which is composed of 14 core tasks (Fig. 11) for now. As shown in Tab. 15, each core task has dozens of instruction prompts (Tab. 15). We will release the pre-processed dataset.

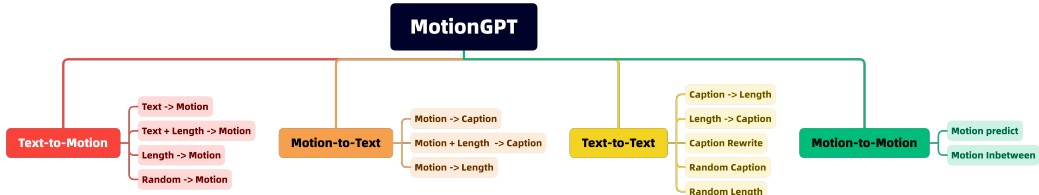

Figure 11: Protocols for multiple motion-language tasks. For each task, we follow Tab. 15 to process the previous datasets [6, 2] into the instruction-based data.

| Task | Input | Output |
|---|---|---|
| Text-to-Motion | Give me a motion that corresponds to [caption]. Demonstrate a sequence of movements that depict [caption]. I need a human motion that conveys [caption]. Can you generate it for me? | [motion] |
| Text-to-Motion w/ length | Give me a motion that lasts for approximately [frames] frames. The caption is: [caption]. Please create a motion that lasts [seconds] seconds and illustrates [caption]. | [motion] |
| Length-to-Motion | Show me a motion that lasts for no more than [frames] frames. Create a motion that has a duration of [seconds] seconds. | [motion] |
| Radnom Motion | Give me motions as you like. Produce actions that are not prescribed. | [motion] |
| Motion-to-Text | Give me a summary of the motion being displayed in [motion] using words. Describe the motion illustrated in [motion] in natural language. | [caption] |
| Motion-to-Text w/ length | Describe the movement portrayed in [motion] that lasts [frames] frames. What is happening in [motion] for a length of [seconds] seconds? | [caption] |
| Motion-to-Length | What is the duration of [motion]'s gestures in frames? What is the total duration of [motion]'s body movements in seconds? | There are [frames] frames in the motion. The motion lasts for [seconds] seconds. |
| Caption-to-Length | How many frames are expected for the motion that matches [caption]? Given [caption], provide the anticipated second duration for the corresponding motion. | The duration is estimated to be around [frames] frames. The motion has a length of [seconds] seconds. |
| Length-to-Caption | What are some possible physical gestures that could be made in [frames] frames? What motion could be performed in [seconds] seconds? | [caption] |
| Random Caption | Depict a motion as like you have seen it. Describe the motion of someone randomly. | [caption] |

Table 15: Some examples of prompt templates in our uniform evaluation protocols.

**Metric Definitions:** We provide more details of evaluation metrics as follows. Our evaluation metrics can roughly divide to five classes including text-motion matching, generation diversity, linguistic quality, motion quality, and time cost. For the first two classes, [17] has already claims clearly and for the linguistic metrics including BLUE [7], Rouge [4], Cider [16], and BertScore [21], you can refer to their own papers for details. Here we focus on the explaination of the rest metrics.

**Motion Quality**. FID, MPJPE, PAMPJPE [1], ACCL have been clearly explained in [17]. Thus here we focus on the Average Displacement Error (ADE) and Final Displacement Error (FDE) refaccuracy of the predicted motion. Following previous motion prediction work[18, 22, 5], ADE is defined as average L2 distance between the ground truth and predicted motion of the whole sequence and FDE is the L2 distance between the ground truth and predicted motion in the last frame.

**Time Costs**. To evaluate the computing efficiency of our models, especially the inference efficiency, we calculate average Frames Per Second (FPS) when generating motions. In our case, we calculate FPS on the test set of HumanML3D [2], set the batch size to one, and ignore the time cost for model and dataset loading parts.

## H  Details on MotionGPT Models

### H.1  Implementation Details

Besides the MotionGPt with 220M parameters, we implement a smaller model that reduces the model dimension with $d_{model} = 512$, $d_{ff} = 2048$ with only 6 layers in encoder and decoder, as well as a larger model with 770 million parameters, which increases the model dimensions with $d_{model} = 1024$, $d_{ff} = 4096$, $d_{kv} = 64$, 24 layers for each transformer. Except for the training iterations during the instruction tuning stage, the other settings are the same. Please refer to Tab. 16 for more details.

| MotionGPT | Small | Base | Large |
|---|---|---|---|
| Backbone | Flan-T5-Small | Flan-T5-Base | Flan-T5-Large |
| Training Batch Size | 64 | 16 | 4 |
| Model Size | 60M | 220M | 770M |
| Pre-training - Iterations | 300K | 300K | 300K |
| Pre-training - Learning Rate | 2e-4 | 2e-4 | 2e-4 |
| Instruction Tuning - Iterations | 200K | 300K | 400K |
| Instruction Tuning - Learning Rate | 1e-4 | 1e-4 | 1e-4 |
| Motion Vocabulary Number $V_m$ | 512 | 512 | 512 |
| Motion Codebook Dimension | 512 | 512 | 512 |

Table 16: Hyperparameters for different MotionGPTs. We train these models on 64 Tesla V100 GPUs. The smaller model size lowers the computational requirements and thus provides faster inference ($cf$. Sec. C). According to Tab. 6, the base MotionGPT model is the best one for overall tasks. However, we believe this could be caused by the small amount of current motion datasets. The large model could achieve the best performance when the amount of data comes up to millions or even billions.