# OpenReview forum: "MotionGPT: Human Motion as a Foreign Language"
_NeurIPS.cc/2023/Conference — NeurIPS 2023 poster_

### Official Review · Reviewer_hFS6 · 2023-07-04

**Soundness:** 3 good
**Presentation:** 3 good
**Contribution:** 3 good
**Rating:** 6
**Confidence:** 3

**Summary:**

This paper proposes MotionGPT, an approach that unifies language modeling with human motion modeling by treating each new motion token the same as a language token. To achieve this, a motion tokenizer based on VQ-VAE is first learned. Then, a pretrained language model is fine-tuned to learn from a unified vocabulary of motion and language and then instruction-finetuned to perform tasks such as text-to-motion, motion completion, and motion captioning. Experiments show that the proposed model achieves state-of-the-art performance for the proposed tasks.

**Strengths:**

- The idea of treating motion as discrete tokens for language models is novel and sound, and can enable a large number of possible applications. The showcased tasks are also comprehensive and demonstrate the flexibility of the proposed model.
- The provided quantitative and qualitative results show state-of-the-art performance and generate natural and good-looking human motion. The multi-task ability is impressive and shows that the model can handle multiple different tasks and natural language instructions.
- Extensive ablation is provided.

**Weaknesses:**

- I do find the lack of failure analysis a bit concerning. It is difficult to gauge how well the model learns the relationship between motion and language.
- Similarly, I find the evaluated motion in the provided video and supplement to be relatively simple and clear instructions, which do not really conform to the objective of "generalizing effectively to unseen tasks or data". For instance, in the text-to-text demonstration, the word "praying" is showcased, but I wonder how well the motion generator part can handle words that have semantic meaning but could be unseen.



Minor: Error T2M-GPT and MDM’s citation are mixed in Table 1.

**Questions:**

Do TM2T, T2M, and poseGPT capture all human motion in their training dataset's discrete latent code? How is the reconstruction loss on ALL the training data?


---

After rebuttal, my main concerns about failure analysis and text complexity are addressed and I would like to maintain a positive rating of this work.

---

**Limitations:**

More failure cases should be discussed.

---

> ### Author Rebuttal · Authors · 2023-08-10
>
> We appreciate your approval of our idea, human motion as a foreign language, as well as what it could enable on applications. We will fix the mixed citations, add more failure cases, and analyze the zero-shot ability of MotionGPT in the paper.
>
> 📝&#8194;**Q:  Failure analysis. Zero-shot ability on handling words that have semantic meaning but could be unseen.**
>
> 💡&#8194;**A:**  As shown in Fig. 12, we provide both zero-shot cases and failure cases. Benefitting from strong language models, MotionGPTs can understand unseen works in the text-to-motion training set, like "scuttling" and "barriers", and generate correct motions based on the meaning of sentences. However, it still struggles to generate unseen motions, like gymnastics, even if MotionGPTs understand the text inputs.
>
> 📝&#8194;**Q: How well MotionGPT learns the relationship between motion and language?**
>
> 💡&#8194;**A:** Unlike the previous motion generators using the text encoder of CLIP for conditions, please note that MotionGPTs leverage language models to learn the motion-language relationship, instead of relying on text features from CLIP. According to our zero-shot results (cf. Fig. 12) and performances on multi-tasks (cf. Fig. 10), MotionGPTs establish robust connections between simple/complex texts and simple motions in evaluations, but they fall short when it comes to complex-text to complex motion translation.
>
>
> 📝&#8194;**Q:  Do TM2T, T2M, and poseGPT capture all human motion in their training dataset's discrete latent code?**
>
> | Method | MPJPE$\downarrow$ | MPJPE $\downarrow$ | ACCL $\downarrow$ | FID $\downarrow$ | DIV $\rightarrow$ |
> |---|---|---|---|---|---|
> | VPoser-t | 75.6 | 48.6 | 9.3 | 1.430 | 8.336 |
> | ACTOR | 65.3 | 41.0 | **7.0** | 0.341 | **9.569** |
> | MLD-1 | **54.4** | 41.6 | 8.3 | 0.247 | 9.630 |
> | MotionGPT (Ours) | 55.8 | **40.1** | 7.5 | **0.067** | 9.675 |
>
> **Motion reconstruciton comparision.**
>
> | Method    | FID $\downarrow$ |
> |-----------|-------------------|
> | MotionGPT (Ours) | $0.510^{\pm.016}$ |
> | T2M-GPT   | $0.514^{\pm.029}$ |
> | MLD       | $\boldsymbol{0.404}^{\pm.027}$ |
>
> **Comparison of FID in text-to-motion task on KIT-ML dataset.**
>
>
> 💡&#8194;**A:**  Given sufficient training or testing data from the same dataset, motion reconstruction is not a challenging task for both VAE and VQ-VAE. We have provided the evaluation on motion reconstruction in Tab.8. However, when dealing with a limited amount of motion data, like the KIT dataset, the VAE model shows better ability in motion interpolation, surpassing VQ-VAE.
> A relevant evaluation is shown above (also in Tab.7), where MLD (VAE) outperforms MotionGPT and T2M-GPT (VQ-VAEs) on FID.
> The real challenge lies in reconstructing complex motions, such as diving or gymnastics sports. Existing motion generators struggle to accurately reconstruct complex motions using a codebook extracted from daily motion datasets. Collecting these complex yet valuable motions is still a significant challenge to the motion research community.

---

> > ### Comment · Reviewer_hFS6 · 2023-08-16
> > **Reviewer Response**
> >
> > I thank the authors for the detailed response.
> >
> > I find most of my concerns resolved and would maintain a positive score.

---

### Official Review · Reviewer_symu · 2023-07-05

**Soundness:** 2 fair
**Presentation:** 3 good
**Contribution:** 3 good
**Rating:** 5
**Confidence:** 4

**Summary:**

In view of the idea that motion could be perceived as a form of body language, the authors propose to fuse motion and language to perform a unified motion-language pre-training.
In detail, motion is quantized into discrete tokens in the same form as natural languages.
Then, language modelling is performed on both motion and text in a unified manner.
Furthermore, prompt tuning is adopted to fine-tune the pre-trained model.
Experiments demonstrate the impressive performance on multiple motion tasks.

**Strengths:**

The idea of unifying motion and language into tokens for uniform pre-training is interesting and novel.

The uniform motion-language model manages to provide a solution for a wide range of tasks.

The provided demo is rather impressive and convincing.

Extensive ablation studies provide a detailed analysis on the effectiveness of different design choices.


**Weaknesses:**

Though the quantized representation provides the ability to unify motion and text, it also imposes constraints on the motion representation due to the sequence-level encoding. In other words, when the operation granularity is smaller than the down-sample rate, I'm not sure whether the method could provide satisfying performance. For example, motion in-between seems to be designed as a token-level in-between. If only given a start frame and an end frame as the in-between input (which could be a more practical application scenario), would the model perform well?

Both HumanML3D and KIT are limited in the vocabulary size and the overall dataset size compared to language datasets as I know. Therefore, I understand the limited performance on KIT and when increasing the scale of the model. While in view of the recent success of LLMs, I think the authors should pay attention to unifying current available datasets to exploit the scalable potential of language models when processing large scale data besides increasing model size.




**Questions:**

I'm interested in the vocabulary that VQ-VAE learned. Is it possible to visualize some of the tokens? Or directly generate description on each single token?

How is the down-sample rate chosen? It is a fundamental hyper-parameter that decides the overall granularity of the model.

**Limitations:**

The authors provide their discussion on the limitation of the paper.

---

> ### Author Rebuttal · Authors · 2023-08-10
>
> 📝&#8194;**Q:  Motion Down-sample, if only given a start frame and an end frame as the in-between input, would the model perform well?**
>
> 💡&#8194;**A:**  VQ-based methods, such as MotionGPT and T2M-GPT, employ downsampling tricky to enhance the density of the codebook or tokens and reduce computing costs. This indeed becomes a constraint when the operation granularity is smaller than the down-sample rate. However, to address this issue, only the start and end frames are provided as in-between inputs. Some technical tricks can be used, such as repeating a single start or end frame up to the window size as inputs and removing the redundant parts in outputs. This does not significantly impact the effectiveness of the model, as there are often static beginnings or endings in the GT motion data.
>
> 📝&#8194;**Q: While in view of the recent success of LLMs, the authors should pay attention to unifying current available datasets to exploit the scalable potential of language models when processing large-scale data besides increasing model size.**
>
> 💡&#8194;**A:**  We appreciate your insight and totally agree with this suggestion. We have faced this limited dataset issue while implementing MotionGPT and in our further research. It is a hard but valuable work to unify and collect a larger motion dataset. Foruthertaly, some researchers are working on this problem, as seen in recent work like Motion-X and other datasets, which hold promise for advancing large-scale motion models. We intend to further evaluate MotionGPT on these larger datasets once they become available.
>
> 📝&#8194;**Q: Visualize some of the tokens in the vocabulary that VQ-VAE learned.**
>
> 💡&#8194;**A:** As shown in Fig.13, we visualize these motion tokens in motion vocabulary $V_m$ and their corresponding localized spatial-temporal contexts, depicted within 4-frame motion segments. However, MotionGPT falls short in generating descriptions for each individual token, as the training is conducted on token sequences.
>
> 📝&#8194;**Q: How is the down-sample rate chosen? It is a fundamental hyper-parameter that decides the overall granularity of the model.**
> | Downsampling | MPJPE $\downarrow$ | MPJPE $\downarrow$ | ACCL $\downarrow$ | FID $\downarrow$ | DIV $\rightarrow$ |
> |---|---|---|---|---|---|
> | $l=1$ | 76.2 | 49.5 | 19.5 | 0.421 | 9.613 |
> | $l=2$ | **52.6** | **37.7** | **9.5** | 0.135 | 9.722 |
> | $l=4$ | 55.8 | 40.1 | 7.5 | **0.067** | 9.675 |
> | $l=8$ | 62.7 | 45.3 | 8.7 | 0.223 | **9.584** |
>
> 💡&#8194;**A:** We selected the down-sample rate based on the frames-per-second (FPS) of the HumanML3D and KIT-ML datasets, which is 20 fps. Therefore, down-sampling by a factor of 4 to achieve 5 fps can ensure distinctiveness in motion frames, and prevents redundancy, and acceleration training. This choice was also made to ensure a fair comparison, as we utilized the same down-sample rate as T2M-GPT. As shown in the above table, we provide an ablation study on these parameters, where a factor of 4 achieves the best FID in motion reconstructions.

---

> > ### Comment · Reviewer_symu · 2023-08-19
> >
> > Thanks for the helpful responses. My major concerns are addressed,

---

### Official Review · Reviewer_gizA · 2023-07-06

**Soundness:** 3 good
**Presentation:** 3 good
**Contribution:** 3 good
**Rating:** 5
**Confidence:** 4

**Summary:**

This paper presents a motion-language model via a shared vocabulary, where the texts are represented by original tokens, and the motions are encoded by a trained discrete tokenizer. Based on a pre-trained encoder-decoder framework, i.e., T5, the authors fine-tune the T5 with masked modeling on motion-language paired data. Finally, the obtained model is finetuned with specific instructions for the target job (text-to-motion, motion-to-text, motion prediction, motion in-between in the paper). The experiments demonstrate the MotionGPT does such tasks well.

**Strengths:**

- this paper has a clear and interesting motivation, i.e., jointly learn motion-language on a token-to-token model, enabling the trained model to be aligned with the text instructions.
- the main paper, along with the supp., provides solid and comprehensive experimental results.

**Weaknesses:**

- It seems the work is finished in a rush. For example, the instruction tuning is performed on each task independently, acting like a simple finetuning (the experiments also give some pieces of evidence, where it depends on the task-specific tuning). In this way, is it just a pretrain-finetune scheme? It is interesting to see any difference with the previous pretrain-finetune pipeline, such as enabling more abilities like reasoning, and zero/few-shot learning.
- The motion prediction lacks comparison with border models like T2M-GPT, since the auto-regressive models are better at prediction.
- Although the authors use a strong pre-trained language model, and finetune it on each task independently, it shows a limited performance gain compared with previous works.

**Questions:**

- Why do you choose T5 as the base model? which is an encoder-decoder architecture. Have you tried a decoder-only model like LLaMA since it is a more straightforward solution for instruction tuning?
- How do you implement the MDM on the motion prediction and in-between tasks? the numbers of MDM exist a large gap with text-to-motion. I mean, for diffusion models, there are lots of details in implementation. For example, reset the prefix 20% motion at each diffusion step. Lacking the details makes the numbers not fully convincing.
- How do you merge the text vocab and motion vocab in detail? concatenating them together?
- For tuning on each task, do you tune the entire model or just part of it?
Table 2 is redundant since all the information is duplicated in the latter tables.

**Limitations:**

Yes

---

> ### Author Rebuttal · Authors · 2023-08-10
>
> Thanks for your approval and insightful comments. We will address your concerns in the following comments, re-organize this redundant Tab. 2, and update our paper accordingly.
>
> 📝&#8194;**Q: Instruction tuning, Reasoning, and Zero-shot learning**
>
> 💡&#8194;**A:** We propose instruction tuning to train a single MotionGPT across all motion-related tasks, while task-specific tuning is to train and evaluate MotionGPTs on a single task. We employ these two training schemes to study the ability of MotionGPT across multi-tasks. As shown in Fig. 12, we provide zero-shot cases. Benefitting from strong language models, MotionGPTs can understand unseen works in the text-to-motion training set, like "scuttling" and "barriers", and generate correct motions based on the meaning of sentences. However, it still struggles to generate unseen motions, like gymnastics, even if MotionGPTs understand the text inputs. Moreover, this reasoning provides inspired insight for our future research. We will explore this direction and provide more detailed zero-shot learning evaluations.
>
> 📝&#8194;**Q: Comparison of motion prediction with T2M-GPT**:
>
>   |Method  | FID $\downarrow$   |Diversity  $\rightarrow$   | ADE $\downarrow$  | FDE  $\downarrow$|
> |:--:|:--:|:--:|:--:|:--:|
>   |Real   |0.002   |9.503  |  -  | - |
>   |MDM   |6.031   |7.813  |5.446  |8.561  |
>   |T2M-GPT  | 2.056 | 8.635|6.161| 8.302 |
>   |MotionGPT (Ours)   | **0.905** | **8.972** | **4.745** | **6.040** |
>
> 💡&#8194;**A:**  We have added T2M-GPT to this comparison of motion prediction in Tab. 5. As shown in the above table, all three methods, MDM, T2M-GPT, and MotionGPTs, follow the same setting with the first 20% and generate the remaining, as mentioned in the implementation details, appendix B.5. Benefitting from larger model size, MotionGPT outperforms two other methods in this task.
>
> 📝&#8194;**Q: Limited performance gain with strong language models.**
>
> 💡&#8194;**A:** We thought MotionGPT, using a significantly larger language model, would surpass all existing methods in all tasks. However, the evaluation shows MotionGPT achieves SOTA results in 18 out of 23 metrics, where many improvements are only small gains. This can be attributed to the limited size of the dataset. As mentioned in R3, both HumanML3D (14,616 motions) and KIT (3,911 motions) are limited in vocabulary size and overall dataset size, particularly when compared to billion-level language datasets, which affects the efficacy of large-scale models. Benefitting from recent dataset works, like Motion-X, we will evaluate the performance gain of MotionGPT in larger datasets once they become available.
>
>
> 📝&#8194;**Q: Why choose T5 as the base model? an encoder-decoder architecture. Have you tried a decoder-only model like LLaMA?**
>
> 💡&#8194;**A:**  The first language model that we used to build MotionGPTs is LLaMA-13B. However, it shows insufficient performance and low training efficiency. We assume the reason is the limited dataset size compared to the large parameters and language data of LLaMA. **As shown in Tab. 15**, we tried a smaller size decoder-only backbone GPT2-Medium and provide the results. Then, we thus choose T5-770M, a small but common language model, as our final backbone, because many previous vision-language multimodal works, like Unified-IO and BLIP, have chosen T5, this encoder-decoder architecture. It shows a strong power to address multi-modal tasks. In addition, the decoder-only model has the advantage for self-supervised without pair data while we have paired data which this advance is greatly weakened. We are still working on collecting a large motion dataset for larger motion-language models.
>
> 📝&#8194;**Q: How do you implement the MDM on the motion prediction and in-between tasks?**
>
> 💡&#8194;**A:**  Thank you for your inquiry. We follow the approach outlined in Appendix B.4 and Line-296 of our paper, where we highlight that MDM achieves the motion in-between task using a masked motion "in-painting" technique. Specifically, this involves fixing the initial and final portions of the motion and allowing the model to generate the central portion. To adapt this concept for motion prediction, we similarly fix a portion of the motion – in our case, the first 20% – and generate the subsequent sequence.
>
> 📝&#8194;**Q: How do you merge the text vocab and motion vocab in detail? concatenating them together?**
>
> 💡&#8194;**A:**  To ensure a shared distribution between language and motion, we initialize the Motion tokens separately and concatenate them alongside the language tokens. This step ensures a balanced representation that encompasses both modalities. Besides the token embeddings are actively trained during the entirety of stages 2 and 3, ensuring a comprehensive fusion of language and motion knowledge. We will also elaborate on this concatenation in the final version.
>
> 📝&#8194;**Q: For tuning on each task, do you tune the entire model or just part of it?**
>
> 💡&#8194;**A:**  To address individual tasks, we adopt a focused approach where the entire model is fine-tuned. Our rationale lies in the fact that, for each specific task, our emphasis is on optimizing task-specific performance, without retaining an excessive amount of intelligence learned from other tasks. Besides, we only exclusively fine-tune the Text-to-Motion task, while other tasks are reported without specific tuning.

---

> > ### Comment · Reviewer_gizA · 2023-08-22
> >
> > Thank you for the detailed responses. I stick to the positive recommendation.

---

### Official Review · Reviewer_rpBK · 2023-07-06

**Soundness:** 3 good
**Presentation:** 3 good
**Contribution:** 3 good
**Rating:** 6
**Confidence:** 5

**Summary:**

This paper introduces a motion generation pipeline called MotionGPT, which is based on GPT. MotionGPT utilizes VQ-VAE to discretize human poses into tokens and combines them with language tokens to create a unified codebook. The model is initially pre-trained on motion language data and subsequently fine-tuned on prompt-based tasks to enable it to perform various motion-language tasks.

**Strengths:**

1. This is the first work that explores the application of Large Language Models (LLMs) in the field of text-driven motion generation. The proposed prompt finetuning method further extends the scope of applications by including 10 different tasks. These methods provide inspiration for future research in this area.

2. The performance in the Motion-to-Text task shows a significant improvement  on Bleu@4 and Cider compared to TM2T.

3. The paper is well-written and effectively conveys information in a clear and understandable manner.

**Weaknesses:**

1. MotionGPT exhibits poorer performance in the crucial text-to-motion task, with a significant gap in FID metrics compared to T2M-GPT. Particularly on the KIT-ML dataset, there is a considerable difference in R Precision compared to T2M-GPT, MLD, and MotionDiffuse.

2. The demo video provides limited comparisons with other examples. Apart from the "crouch down" example, the performance of MotionGPT is not noticeably superior to T2M-GPT in the provided examples. These examples do not sufficiently demonstrate an advantage in terms of generation quality.

3. The overall technical contribution is limited. The major distinction from T2M-GPT lies in the combination of motion modality and language modality for modeling, along with subsequent prompt finetuning. However, in the current version, they appear more like shared intermediate layers rather than truly integrated. For instance, a potential approach could involve language-based motion editing, where given a reference motion sequence and a desired text modification, the algorithm produces the edited result. Such task types would better illustrate the advantages of unified modeling. Additionally, considering the significant performance gap between MotionGPT and T2M-GPT in standard text-to-motion tasks, MotionGPT seems to sacrifice accuracy in exchange for additional functionalities. Such technical contribution does not meet the bar set by NeurIPS conference.

**Questions:**

1. Can MotionGPT perform motion editing or motion composition similar to MotionDiffuse and MDM?

2. The supplementary material states that there were only 15 users in the user study for Motion-to-Text. The number of testers for Text-to-Motion is not mentioned. However, 15 users may be considered an insufficient sample size for a reliable evaluation, especially for Motion-to-Text. It is recommended to have a larger sample size, preferably around 50 or more, to provide a more credible assessment.

3. What is the reason behind the significant difference in performance between the KIT-ML and HumanML3D datasets?

**Limitations:**

Limitations have been well discussed.

---

> ### Author Rebuttal · Authors · 2023-08-10
>
> 📝&#8194;**Q: Motion Quality and Performance Gain**
>
> | Method    | FID $\downarrow$ |
> |:--|:--|
> | MDM  | $0.544^{\pm.044}$ |
> | MotionGPT | $0.160^{\pm.008}$ |
> | T2M-GPT   | $\boldsymbol{0.116}^{\pm.004}$ |
>
> Comparison of FID in text-to-motion task on HumanML3D dataset.
>
> | Method    | FID $\downarrow$  |
> |:--|:--|
> | T2M-GPT   | $0.514^{\pm.029}$ |
> | MotionGPT | $0.510^{\pm.016}$ |
> | MDM       | $\boldsymbol{0.497}^{\pm.021}$ |
>
> Comparison of FID in text-to-motion task on KIT-ML dataset.
>
> 💡&#8194;**A:**  The FID metrics primarily focuses on the motion quality rather than the correlation between motion and text. While MDM serves as a successful benchmark for motion generation, both MotionGPT and T2M-GPT outperform MDM by a margin of 0.38~0.43 on the FID scale. However, the difference in motion quality among these three works is not significant in video supply. Additionally, MDM outperforms two vector quantized methods, MotionGPT and T2M-GPT, in terms of FID on the KIT dataset. This can be attributed to the limited number of 3,911 motion sequences, which makes it challenging to construct a comprehensive motion codebook. More importantly, MotionGPT contributes to multiple motion tasks with LLM, particularly in generating both text and motion within a single model, rather than aiming to improve the FID metric.
>
>
> 📝&#8194;**Q: Performance Gain on R-Precision in KIT**
> 💡&#8194;**A:** The evaluation of R-Precision in the KIT dataset relies on the text encoder, which is built using a limited set of 6,353 textual descriptions. In contrast, MotionGPTs benefit from LLM and large language data, enabling them to generate longer and more nature language descriptions for motion. However, this leads to a discrepancy between the generated descriptions and the GT descriptions, resulting in a lower R-Precision.
>
> 📝&#8194;**Q: MotionGPT seems to sacrifice accuracy in exchange for additional functionalities.**
>
> 💡&#8194;**A:** As shown in Fig. 10, MotionGPT achieves SOTA on 18 out of 23 metrics across four motion-related tasks. Additionally, as mentioned by R3, both HumanML3D and KIT are limited in overall dataset size, particularly when compared to billion-level language datasets. This affects the efficacy of large-scale models. We will further employ a larger motion-text dataset to evaluate MotionGPT. Besides, MotionGPTs introduce motion-language pre-training, as well as its zero-shot ability, which is a promising direction worth exploring and could stimulate self-training procedures for further research.
>
> 📝&#8194;**Q: Can MotionGPT perform motion editing or motion composition similar to MotionDiffuse and MDM?**
>
>   |Method  | FID $\downarrow$   |DIV  $\rightarrow$   | ADE $\downarrow$  | FDE  $\downarrow$|
> |:--|:--|:--|:--|:--|
>   |Real   |0.002   |9.503  |  -  | - |
>   |MDM   |6.031   |7.813  |5.446  |8.561  |
>   |T2M-GPT  | 2.056 | 8.635|6.161| 8.302 |
>   |**MotionGPT (Ours)**   | **0.905** | **8.972** | **4.745** | **6.040** |
>
> Comparison of motion prediction on HumanML3D dataset using motion data only.
>
> 💡&#8194;**A:**  Referring to MDM, motion editing has two categories: body part editing and motion completion in the temporal domain. MotionGPT is capable of the latter, which includes motion prediction and motion in-between. It outperforms both MDM and T2M-GPT in table above. However, when it comes to body part editing, the vector quantization(VQ)-based methods, like MotionGPT and T2M-GPT, are not as suitable as diffusion-based models that utilize diffusion inpainting on raw motion data. We agree that editing body parts with LLM and prompts is a promising direction but still needs exploration.
>
> 📝&#8194;**Q: Technical contribution.**
>
> 💡&#8194;**A:**  Thanks for pointing out that MotionGPT is the first work that explores motion generation with LLM. Firstly, we propose this motion-language pre-training on LLM rather than using CLIP models like previous methods. It is not a trivial combination since it needs to model and generate two distinct modes from scratch. To achieve this on MotionGPTs, we thus introduce a new training procedure: a motion-language pre-training stage and an instruction tunning stage. Secondly, we propose MotionGPT as a uniform motion-language generative pre-trained model to address various motion tasks, particularly in text-to-motion and motion-to-text. To our best knowledge, it is the first exploration of achieving such large models in the motion domain. We have developed new instruction templates and multi-task evaluation protocols, which could also contribute to the motion domain.
>
> 📝&#8194;**Q: User-study**
>
> 💡&#8194;**A:**  We achieve a more detailed user study to evaluate our model's performance. For text-to-motion assessment, we generated motions for 80 HumanML3D test set descriptions, comparing MotionGPTs with MDM and T2M-GPT, alongside GT. Semantic and realism studies presented text-video pairs to participants, asking which motion **corresponded better** or was **more realistic**, respectively. In the motion-to-text study, we visualized 50 GT motions with GT descriptions and generated corresponding textual descriptions using TM2T and our method. Each participant addressed a batch of questions randomly from all questions, and 19 unqualified participants among a total of 110 samples were identified and excluded by 2 'catch trials' questions. Each video pairs were reviewed by multiple participants, with a majority vote determining superior methods. Equal scores were assigned for tied results.
>
> |  Question | MotionGPT vs MDM | MotionGPT vs T2M-GPT | MotionGPT  vs GT |
> |:--|:--:|:--:|:--:|
> | Which of the two motions is more realistic? | 54% | 53% | 48% |
> | Which of the two motions corresponds better to the text prompt? | 57% | 56% | 49% |
>
> | Question | MotionGPT vs GT | MotionGPT vs TM2T |
> |:--|:--:|:--:|
> | Which description can better describe the motion? | 48% | 55% |
>
> The results above indicate improved action quality alignment with text and motion, similar to ground truth.

---

> > ### Comment · Reviewer_rpBK · 2023-08-17
> > **Post-rebuttal Comment**
> >
> > I express my gratitude to the authors for their comprehensive rebuttal. I am pleased to note that my initial concerns have been satisfactorily addressed. I have also taken into account the input from other reviewers, and it appears that no significant additional concerns have been raised. I am inclined toward recommending acceptance and will revise my score after reviewer discussion period.

---

> > > ### Author Response · Authors · 2023-08-21
> > >
> > > We sincerely appreciate the recognition of our work. Your valuable insights are greatly appreciated. Additional evaluation/ablation and corresponding explanations will be included in the final version.

---

### Author Rebuttal · Authors · 2023-08-10

We thank all the reviewers for approvals: The idea of **unifying motion and language into tokens for uniform pre-training** is **novel and sound** (R3, R4), and this motivation is **clear and interesting** (R2). This paper provides **inspiration for future research** (R1) and **impressive demo** (R3); has **comprehensive experimental results** (R2), **extensive ablation studies** (R3, R4), and **impressive multi-task ability** (R4). We will address the concerns and fix the mixed citations.
(Reviewer rpBK - R1, Reviewer gizA - R2, Reviewer symu - R3, Reviewer hFS6 - R4)

**Motivation and Novelty** :
We present MotionGPT to address various human motion-related tasks within one single unified model, by unifying motion modeling with language through a shared vocabulary. To train this unified model, we propose an instructional training scheme under the protocols for multiple motion-language tasks, which further reveals the potential of Large Language Models (LLMs) in motion tasks beyond the success of language generation. However, it is non-trivial for this combination since it needs to model and generate two distinct modes from scratch. Contrary to the previous work leveraging CLIP to extract text embedding as motion generation conditions, like T2M-GPT, MotionGPT introduces the motion-language pre-training on LLM so it can leverage the strong language generation and zero-shot transfer abilities (See Fig.12) of pre-trained language models, as well as generates human language and motion in a unified model.

**Limited Datasets and Evaluation Metrics** : Both HumanML3D (14,616 motions) and KIT (3,911 motions) are limited in the vocabulary size and the overall dataset size, also mentioned by Reviewer symu, particularly when compared to billion-level language datasets. This hampers the efficacy of large-scale models within the motion domain. The KIT dataset, with only 3,911 motion sequences, falls short in training large models with billions of parameters, as the extracted motion vocabulary struggles to represent all potential motions. Fortunately, during this review, recent works such as Motion-X, propose significantly larger motion datasets with multi-modal annotations, which hold promise for advancing large-scale motion models. We intend to further evaluate MotionGPT on these larger datasets once they become available.

Furthermore, the metrics employed for motion-to-text evaluation, such as R-Precision in KIT, are dependent on the text encoder, which is constructed from a limited pool of 6,353 textual descriptions. In contrast, MotionGPTs benefit from LLM and large language data, enabling them to generate longer and more nature language descriptions for motion. However, this leads to a discrepancy between the generated descriptions and the ground truth (GT) descriptions, resulting in a lower R-Precision. We also notice a recent work, Text-to-Motion Retrieval (TMR), a contrastive model on text and motion, which could assist MotionGPT in more accurate text-motion evaluations.

---

### Decision · Program_Chairs · 2023-09-21

**Decision:**

Accept (poster)

**Comment:**

This paper was reviewed by four experts in the field. Based on the reviewers' feedback, the decision is to recommend the paper for acceptance to NeurIPS 2023. The reviewers did raise some valuable concerns that should be addressed in the final camera-ready version of the paper. The authors are encouraged to make the necessary changes to the best of their ability. We congratulate the authors on the acceptance of their paper!